# Easy Learning from Label Proportions

**Robert Busa-Fekete**
Google Research
busarobi@google.com

**Heejin Choi**[*]
Coupang Inc
hechoi53@coupang.com

**Travis Dick**
Google Research
tdick@google.com

**Claudio Gentile**
Google Research
cgentilek@google.com

**Andres Munoz Medina**
Google Research
ammmedina@google.com

## Abstract

We consider the problem of Learning from Label Proportions (LLP), a weakly supervised classification setup where instances are grouped into i.i.d. "bags", and only the frequency of class labels at each bag is available. Albeit, the objective of the learner is to achieve low task loss at an individual instance level. Here we propose EASYLLP, a flexible and simple-to-implement debiasing approach based on aggregate labels, which operates on arbitrary loss functions. Our technique allows us to accurately estimate the expected loss of an arbitrary model at an individual level. We elucidate the differences between our method and standard methods based on label proportion matching, in terms of applicability and optimality conditions. We showcase the flexibility of our approach compared to alternatives by applying our method to popular learning frameworks, like Empirical Risk Minimization (ERM) and Stochastic Gradient Descent (SGD) with provable guarantees on instance level performance. Finally, we validate our theoretical results on multiple datasets, empirically illustrating the conditions under which our algorithm is expected to perform better or worse than previous LLP approaches.

## 1 Introduction

In traditional supervised learning problems, a learner has access to a sample of labeled examples. This collection of labeled examples is used to fit a model – to name a few, decision trees, neural networks, random forests – by minimizing a loss over the observed sample. By contrast, in the problem of Learning from Label Proportions (LLP), the learner only observes collections of unlabeled feature vectors called *bags*, together with the proportion of positive examples in each bag. The LLP problem is motivated by a number of applications where access to individual examples is too expensive or impossible to achieve, or available at aggregate level for privacy-preserving reasons. Examples include e-commerce, fraud detection, medical databases [20], high energy physics [6], election prediction [30], medical image analysis [4], remote sensing [7].

As a weakly supervised learning paradigm, LLP traces back to at least [5, 19, 22, 23, 32], and was motivated there by learning scenarios where access to individual examples is often not available. A paradigmatic example is perhaps a political campaign trying to predict the preference of the electorate. Since voting is anonymous, a political analyst may not be able to observe individual votes, yet, they have access to aggregate voting preferences at the district level.

The problem has received renewed interest more recently (e.g., [8, 17, 26, 24, 25, 31, 16, 34]), driven by the desire to provide more privacy to user information. For instance the ad conversion reporting

---

[*]Work done while at Google.

37th Conference on Neural Information Processing Systems (NeurIPS 2023).

system proposed by Apple, SKAN, allows an ad tech provider to receive information about the conversions (e.g., purchases) from customers only aggregated across multiple impressions. This aggregation is intended to obfuscate the individual customer's activity. A similar API has also been proposed by Google Chrome and Android to report aggregate conversion information (e.g., [1]). Given the importance of conversion modeling for online advertising, learning how to train a model using only these aggregates has become a crucial task for many data-intensive online businesses.

Research in LLP can be coarsely divided into two types of goals: Learning a bag classifier that correctly predicts label proportions, and learning an individual classifier that can correctly predict instance labels. The former has been the focus of most of the literature in this area. Representative papers include [5, 19, 32, 33]. While training classifiers to match label proportions is an obvious heuristic, little work has been done in trying to understand under which conditions these classifiers would be accurate when generating event level predictions. Yu et al. [33] provide guarantees that suggest that, under the absolute loss, predicting label predictions to very high accuracy can result in good predictions at event level. On the other side of this research area, finding a good instance level predictor using only label proportions has so far remained elusive and under-explored in its generality. The solutions introduced so far require either making some assumptions on the data generation process [26, 34] or on the model class [22]. Other solutions involve solving complex combinatorial problems [8] or require that an example belongs to multiple bags [24, 25].

In this work we provide a two pronged approach towards making LLP on *randomly generated* bags seamless. First, we contribute to a better theoretical understanding of a well-known label proportion prediction algorithm. This algorithm simply trains a model whose average prediction on a bag is as close as possible to the label proportion in that bag. This straightforward algorithm is referenced (either explicitly or implicitly) in multiple past works (e.g., [19, 23, 2]), yet is hardly analyzed theoretically, or even considered as a baseline for experimental comparisons. Here, we show under what conditions this folklore algorithm can produce good event level predictions. Moreover, we show through extensive experimentation that when those conditions are met this algorithm seems to outperform previous baselines tailored to generating event level predictions. On the flip side, when such conditions are not met, we also show that the quality of this algorithm quickly deteriorates as the bag size increases.

A more robust and flexible approach to LLP that operates at the instance level is EASYLLP, a reduction method virtually applicable to *any* machine learning task. Unlike many algorithmic proposals in this space, the implementation of EASYLLP requires trivial modifications to current machine learning training pipelines. We elucidate the flexibility of our approach by applying it to two widely interesting algorithmic techniques, Empirical Risk Minimization (ERM) and Stochastic Gradient Descent (SGD). Our findings are complemented by an extensive experimental investigation on a diverse suite of benchmark datasets.

**Main contributions.** The contribution of our paper can be summarized as follows.

1. We provide a theoretical analysis of a popular label proportion matching algorithm, that is suggestive of the conditions under which this algorithm is expected to work in practical scenarios. Despite its relative simplicity, the scope of this analysis is broad and, to the best of our knowledge, original.

2. We provide a general debiasing technique for estimating the expected instance loss (or loss gradient) of an arbitrary model using only label proportions, and thoroughly quantify the variance of this estimator.

3. We provide a reduction of ERM with label proportions to ERM with individual labels, and show that when the learner observes bags of size $k$, the sample complexity of ERM with label proportions increases only by a factor of $k$. Likewise, we provide an analysis of SGD using only label proportions and show that for bags of size $k$, its regret increases by only a factor of $k$.

4. We carry out an extensive set of experiments comparing the proportion matching method and EASYLLP to known methods available in the LLP literature. The experiments are designed to encompass diverse combinations of datasets and learning models. We identify general trends in the relative performance of the tested methods when evaluated at the instance level. To the best of our knowledge, in spite of its folklore status, this is the first thorough investigation that involves the proportion matching algorithm and its comparison to event

level classifiers. We believe the latter is by itself an important contribution on its own, as proportion matching turns out to be a strong baseline in a number of cases.

**Related work.** Interest in LLP traces back to at least [5, 19, 22, 23, 29, 32, 20]. The literature in recent years has become quite voluminous, so we can hardly do justice of it here. In what follows, we comment on and contrast to the references from which we learned about LLP problems.

In [5] the authors consider a hierarchical model that generates labels according to the given label proportions and proposed an MCMC-based inference scheme which, however, is not scalable to large training sets. [19] shows how standard supervised learning algorithms (like SVM and $k$-Nearest Neighbors) can be adapted to LLP by a reformulation of their objective function via label proportion matching. Yet, no experiments are reported on classification tasks. In [22], the authors propose a theoretically-grounded way of estimating the mean of each class through the sample average of each bag, along with the associated label proportion. The authors make similar assumptions to ours, in that the class-conditional distribution of data is independent of the bags. However, their estimators rely on very strong assumptions, like conditional exponential models, which are not a good fit to nowadays Deep Neural Network (DNN) models. Similar limitations are contained in [20]. [23] proposes an adaptation of SVM to the LLP setting through a scheme which can be seen as calibration on top of proportion matching. Yet, this turns out to be restricted to linear models in some feature space. Similar limitations are in the $\alpha$-SVM method proposed in [32], the non-parallel SVM formulation of [21], and the pinball loss SVM in [28]. The original $\alpha$-SVM formulation was extended to other classifiers; e.g., [15] extends the formulation to CNNs with a generative model whose associated maximum likelihood estimator is computed via Expectation Maximization, which turns out not to be scalable to sizeable DNN architectures. [29] proposes a method based on $k$-means to identify a clustering of the data which is compatible with the label proportions, but their method suffers from an extremely high computational complexity.

Many of these papers are in fact purely experimental in nature, and their main goal is to adapt the standard supervised learning formulation to an LLP formulation so as to obtain a bag level predictor.

On the learning theory side, besides the already mentioned [22, 20], are the efforts contained in [24, 25], the task of learning from multiple unlabeled datasets considered in [17, 16], and the statistical learning agenda pursued in [26, 34] (and references therein from the same authors). In [24, 25] the author is essentially restricting to linear-threshold functions and heavily relies on the fact that an example can be part of multiple bags, while we are working with non-overlapping i.i.d. bags and general model classes. In [17, 16] the authors consider a problem akin to LLP. Similar to our paper, the authors propose a debiasing procedure via linear transformations. Yet, the way they solve the debiasing problem forces them to impose further restrictions on the bags, like the separation of the class prior distributions across different bags. It is this diversity that allows the authors to construct unbiased estimates and then derive consistency results. On the contrary, the bags proposed in our setup are drawn i.i.d. from the same prior distribution, a scenario where many of these algorithms would fail. Hence, we work under the assumption that we cannot handcraft diverse bags out of our samples, as the aggregation into bags is done without having access to the class conditional distributions (which [17, 16] and related papers heavily rely upon). Moreover, the convergence results to the event level performance are only proven in those papers with specific families of loss function (e.g., proper loss functions). See Section 5 for further in-context discussion.

The work [26] introduced a principled approach to LLP based on a reduction to learning with label noise. As in [17], their basic strategy is to pair bags, and view each pair as a task of learning with label noise, where label proportions are related to label flipping probabilities. The authors also established generalization error guarantees at the event level, as we do here. [34] extend their results to LLP for multiclass classification. From a technical standpoint, these two papers have similar limitations as [17, 16]. Besides, the risk measure they focus on is balanced risk rather than classification risk, as we do here.

In our experiments (Section 7), we empirically compare EASYLLP to a folklore label proportion matching method, to the MeanMap method from [22], and to a label generation approach from [8], the latter viewed as representative of recent applications of DNNs to LLP.

## 2 Setup and Notation

Let $\mathcal{X}$ denote a feature (or instance) space and $\mathcal{Y} = \{0, 1\}$ be a binary[2] label space. We assume the existence of a joint distribution $\mathcal{D}$ on $\mathcal{X} \times \mathcal{Y}$, and let $p = \mathbb{P}_{(x,y) \sim \mathcal{D}}(y = 1)$ denote the probability of drawing a sample $(x, y) \in \mathcal{X} \times \mathcal{Y}$ from $\mathcal{D}$ with label $y = 1$. For a natural number $n$, let $[n] = \{i \in \mathbb{N} \colon i \leq n\}$.

A labeled *bag* of size $k$ is a sample $\mathcal{B} = \{x_1, \ldots, x_k\}$, together with the associated *label proportion* $\alpha(\mathcal{B}) = \frac{1}{k} \sum_{j=1}^{k} y_j$, where $(x_1, y_1), \ldots, (x_k, y_k)$ are drawn i.i.d. according to $\mathcal{D}$. We assume the learner has access to a collection $\mathcal{S} = \{(\mathcal{B}_i, \alpha_i), i \in [n]\}$ of $n$ labeled bags of size $k$, where $\mathcal{B}_i = \{x_{ij} \colon j \in [k]\}$, $\alpha_i = \alpha(\mathcal{B}_i) = \frac{1}{k} \sum_{j=1}^{k} y_{ij}$ is the label proportion of the $i$-th bag, and all the involved samples $(x_{ij}, y_{ij})$ are drawn i.i.d. from $\mathcal{D}$. In words, the learner receives information about the $nk$ labels $y_{ij}$ of the $nk$ instances $x_{ij}$ only in the aggregate form determined by the $n$ label proportions $\alpha_i$ associated with the $n$ labeled bags $(\mathcal{B}_i, \alpha_i)$ in collection $\mathcal{S}$. Notice, however, that the instances $x_{ij}$ are individually observed.

Given a hypothesis set $\mathcal{H}$ of functions $h$ mapping $\mathcal{X}$ to a prediction space $\widehat{\mathcal{Y}}$, and a loss function $\ell \colon \widehat{\mathcal{Y}} \times \mathcal{Y} \to \mathbb{R}^+$, the learner receives a collection $\mathcal{S}$, and tries to find a hypothesis $h \in \mathcal{H}$ with the smallest *population loss* (or *risk*) $\mathcal{L}(h) = \mathbb{E}_{(x,y) \sim \mathcal{D}}[\ell(h(x), y)]$ with high probability over the random draw of $\mathcal{S}$. When clear from the surrounding context, we will omit subscripts like "$(x,y) \sim \mathcal{D}$" or "$\mathcal{D}$" from probabilities and expectations.

We shall consider two broadly used learning methods for solving the above learning problem, Empirical Risk Minimization (ERM, Section 5), or regularized versions thereof, and Stochastic Gradient Descent (SGD, Section 6). In this latter context, we will consider a parameter space $\mathcal{W}$ and consider a learner that tries to optimize a loss $\ell \colon \mathcal{W} \times \mathcal{X} \times \mathcal{Y} \to \mathbb{R}$ iteratively over a collection of bags. Before that, we find it instructive to delve into the theoretical properties of an approach to LLP which is by now belonging to folklore.

## 3 Proportion Matching Algorithm

We now introduce a simple and very well known algorithm for learning from label proportions. Yu et al. [33] refers to this algorithm as Empirical Proportion Risk Minimization but different versions of the algorithm are discussed in the LLP literature without a clear reference to its origin.

**Definition 3.1.** Given a loss function $\ell \colon \mathbb{R} \times \mathbb{R} \to \mathbb{R}^+$, a hypothesis set of functions $\mathcal{H}$, and a collection $\mathcal{S} = \{(\mathcal{B}_i, \alpha_i), i \in [n]\}$ of $n$ labeled bags of size $k$, the PROPMATCH algorithm minimizes the empirical *proportion matching loss*, i.e. it solves the following optimization problem

$$\min_{h \in \mathcal{H}} \sum_{i=1}^{n} \ell \left( \frac{1}{k} \sum_{j=1}^{k} h(x_{ij}), \alpha_i \right) . \tag{1}$$

That is, the PROPMATCH algorithm creates a bag level prediction by simply averaging the predictions of a model on the bag's individual examples. Yu et al. [33] prove a uniform convergence guarantee implying that, given enough data, the minimizer of the empirical proportion matching loss is an approximate minimizer of the *population level* proportion matching loss:

$$\min_{h \in \mathcal{H}} \mathbb{E}_{\mathcal{B},\alpha} \left[ \ell \left( \frac{1}{k} \sum_{x \in \mathcal{B}} h(x), \alpha \right) \right] . \tag{2}$$

In general, the minimizer of the population level proportion matching loss may not produce accurate event-level predictions. However, Yu et al. [33] further show that when $\ell$ is the absolute loss, achieving small population level proportion matching loss also leads to a classifier with small population level event loss.

We now strengthen those results and show that under some general conditions on the loss function, and if the function class $\mathcal{H}$ is expressive enough, minimizing the population level proportion matching loss is equivalent to minimizing the population level event loss.[3]

---

[2]A multiclass extension of all our results is also possible – see Appendix C.

[3]Due to lack of space, all proofs are given in the appendix.

**Theorem 3.2.** *Assume $\mathcal{H}$ is such that the function $h^*: x \mapsto \mathbb{P}(y=1|x)$ is in $\mathcal{H}$. Let $\ell: \mathbb{R} \times \mathbb{R} \to \mathbb{R}^+$ be such that, for any random variable $Z$, $q = \mathbb{E}[Z]$ is the* unique *solution of $\min_{r \in \mathbb{R}} \mathbb{E}_Z[\ell(r, Z)]$. Then, $h^*$ is a minimizer of (2). Moreover, every other minimizer $h$ satisfies $\mathbb{P}(h(x){=}h^*(x)) = 1$.*

**Corollary 3.3.** *Let $\mathcal{H}$ satisfy the conditions of Theorem 3.2. Then $h^*$ as defined above is the unique minimizer for the proportion matching loss when $\ell$ is the square loss or the binary cross-entropy loss.*

The above corollary provides us with conditions for the proportion matching loss minimizer to also minimize the event level loss. It is natural to ask ourselves, what the behavior of the proportion matching loss would be when those conditions are violated. To our knowledge, this remains an open problem. However, the following example shows that if the model class $\mathcal{H}$ is not expressive enough, then using the proportion matching loss can in fact lead to an arbitrarily bad event level predictor.

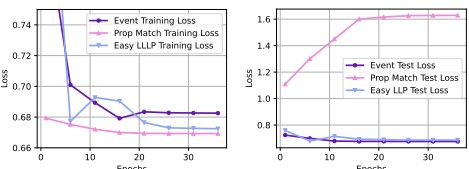

Figure 1: **Left:** Training loss for different learning approaches. PROPMATCH (using (1)), EASYLLP (using (3), Section 4), and event level using regular cross-entropy loss. **Right:** Test loss using event level labels to measure cross-entropy loss for all methods. Notice how the PROPMATCH training loss decreases but the test loss continues to increase. EASYLLP tracks the event level loss as suggested by theory.

Let $\mathcal{X} = \mathbb{R}^2$ and $\mathcal{Y} = \{0, 1\}$. We define a latent variable $\eta$ uniformly distributed in $\{-1, 1\}$. We generate a distribution $\mathcal{D}$ over pairs $(x, y)$ as follows: first sample a latent variable $\eta$, then sample $x$ according to the Gaussian distribution $N((\eta, \eta), I)$, where $I \in \mathbb{R}^{2 \times 2}$ is the identity matrix. Finally, sample $y$ such that $\mathbb{P}(y = 1) = \frac{e}{e+1}$ if $\eta = -1$ and $\frac{e^4}{e^4+1}$ if $\eta = 1$. Using SGD, we attempt to fit a logistic regression model parametrized by $w \in \mathbb{R}^2$ where $\widehat{p}_w(y = 1|x) = \frac{e^{w \cdot x}}{1+e^{w \cdot x}}$. Crucially the hypothesis class used to model $\mathcal{D}$ does not include the true conditional probability distribution. For the experiment we generate 100000 examples from distribution $\mathcal{D}$, and group them randomly into bags of size 500. Figure 1 shows the effect of training a model using proportion matching loss vs. event level loss. Note that even though the empirical proportion matching loss is decreasing, the test loss (on events) increases.

To summarize, we have shown that PROPMATCH can, for the most popular classification and regression losses and under some assumptions on $\mathcal{H}$, recover a very good instance level classifier. Nonetheless, when those assumptions are violated the performance of PROPMATCH can drastically degrade. We believe these observations pave the way for new exciting research in fully understanding this simple algorithm.

## 4 Easy LLP

Based on the results of the previous section, we are interested in defining a robust, theoretically founded algorithm for learning from label proportions. We now introduce the main tool for our approach.

**Definition 4.1.** Let $g: \mathcal{X} \times \mathcal{Y} \to \mathbb{R}^d$ be any (measurable) function, for some output dimension $d \geq 1$. Let also $p = \mathbb{E}[y]$ be the probability of observing a positive label. We define $\widetilde{g}: \mathcal{X} \times [0, 1] \to \mathbb{R}^d$, the *soft-label corrected function* associated with $g$, as

$$\widetilde{g}(x, \alpha) = \big(k(\alpha - p) + p\big)g(x, 1) + \big(k(p - \alpha) + (1 - p)\big)g(x, 0). \tag{3}$$

The main property of $\widetilde{g}$, which is relevant to LLP, is that it is as unbiased estimator of $\mathbb{E}_{(x,y) \sim \mathcal{D}}[g(x, y)]$, as we next show.

**Proposition 4.2.** *Given a sample $(x_1, y_1), \ldots, (x_k, y_k)$ drawn i.i.d. according to $\mathcal{D}$, let $(\mathcal{B}, \alpha)$ be the corresponding labeled bag of size $k$, for some $k \geq 1$. Let $g: \mathcal{X} \times \mathcal{Y} \to \mathbb{R}^d$ be any (measurable) function, for some output dimension $d \geq 1$, and $\widetilde{g}: \mathcal{X} \times [0, 1] \to \mathbb{R}^d$ be its associated soft-label corrected function. Then for every element $x_j \in \mathcal{B}$ we have*

$$\mathbb{E}_{(\mathcal{B}, \alpha)}[\widetilde{g}(x_j, \alpha)] = \mathbb{E}_{(x,y) \sim \mathcal{D}}[g(x, y)].$$

For empirical risk minimization (ERM), we shall apply Proposition 4.2 with the function $g$ that computes the per-example loss of a given model, so that we obtain unbiased estimates of the model's

population level loss. Similarly, for gradient-based optimization, we take $g$ to be the function that computes the per-example gradient of the loss w.r.t. the model parameters so that we obtain an unbiased gradient estimate. More details of our application of Proposition 4.2 are given at the end of this section.

Proposition 4.2 shows that we can easily obtain an unbiased estimate of the expectation of any function $g$ by applying a simple linear transformation to the output of $g(x, y)$, for $y = 0, 1$. We would like to highlight the importance of this simple proposition. Whereas there has been a lot of research in LLP, to our knowledge this is the first expression that shows that one can recover an unbiased estimate of an arbitrary function $g$ using only information from label proportions.

While the above result provides us with a straightforward way to estimate the expectation of a function $g$ (which can for instance be specialized to a loss function), note that the variance of $\widetilde{g}$ increases as the number of elements in each bag grows. Indeed, since all terms in the definition of $\widetilde{g}$ have a factor of $k$, we might expect the variance of the estimator to grow as $k^2$, which could be prohibitively large even for moderate values of $k$. Notice however that because $\alpha = \frac{1}{k} \sum_{j=1}^{k} y_j$, and each $y_j \sim \text{Bernoulli}(p)$, we expect by standard concentration arguments that $k(\alpha - p) \in O(\sqrt{k})$ which should imply that the variance scales like $k$. The following theorem shows that indeed, the variance of these estimates is of order $k$ and not $k^2$. In addition, since $\widetilde{g}(x_j, \alpha)$ is unbiased for all $j \in [k]$, so is $\frac{1}{k} \sum_{j=1}^{k} \widetilde{g}(x_j, \alpha)$. The same theorem also shows that the variance of the latter estimator is always smaller than the variance of the former. That is, using all $k$ samples in a bag is always better (in terms of variance) than using any single sample.

**Theorem 4.3.** *Let $g \colon \mathcal{X} \times \mathcal{Y} \to \mathbb{R}^d$ be such that $\sup_{x, y} \|g(x, y)\|^2 \leq M$, and denote by $\widetilde{g}$ its associated soft labeled corrected function. Also, set for brevity $g_0 = g(x, 0)$ and $g_1 = g(x, 1)$ and, for each $j \in [k]$, $\widetilde{g}_j = \widetilde{g}(x_j, \alpha)$. Then, for any size $k \geq 1$ and any $j \in [k]$,*

$$\mathbb{E}[\|\widetilde{g}_j\|^2] = \mathbb{E}[\|g(x_j, y_j)\|^2] + (k-1)p(1-p)\,\mathbb{E}\left[\|g_0 - g_1\|^2\right]$$

$$\mathbb{E}\left[\left\|\frac{1}{k}\sum_{i=1}^{k}\widetilde{g}_i\right\|^2\right] \leq \mathbb{E}[\|\widetilde{g}_j\|^2]. \tag{4}$$

*Moreover, there exists a universal constant $C$ such that*

$$\mathbb{E}\left[\left\|\frac{1}{k}\sum_{i=1}^{k}\widetilde{g}_i\right\|^2\right] \leq C + kp(1-p)\left\|\mathbb{E}[g_0 - g_1]\right\|^2,$$

*where $p = \mathbb{P}_{(x, y) \sim \mathcal{D}}(y = 1)$.*

The bound in the above theorem confirms our intuition. Moreover, it shows that the variance grows slower for datasets where $p$ is close to $1$ or $0$. This is intuitively clear, for very skewed datasets, we expect label proportions to provide a better description of the true labels. In the extreme cases where $p = 0$ or $p = 1$, LLP becomes equivalent to learning from individual examples.

The results of this section have demonstrated that for any function $g$, one can obtain an estimator of its expectation using only label proportions. More importantly the variance of this estimator only scales linearly with the bag size.

**Note about knowledge of population level positive rate.** At this point the reader is aware that the definition of the soft label corrected function requires knowledge of the population level positive rate $p$. While the exact value of $p$ is unknown, one can easily estimate it from the label proportions itself. Indeed, using the fact that the generated bags are i.i.d. it is easy to see that $\widehat{p} = \frac{1}{n} \sum_{i=1}^{n} \alpha_i = \frac{1}{nk} \sum_{i, j} y_{ij}$ is a very good estimator for $p$.

**EASYLLP.** We now have all elements to introduce the EASYLLP framework for learning from label proportions. The framework consists of specializing the function $g$ for particular learning tasks. Two notable instantiations of EASYLLP which we will analyze in further sections are ERM and stochastic gradient descent (SGD). For ERM, given a hypothesis $h$ and a loss function, we let $g_h(x, y) = \ell(h(x), y)$ and the corresponding soft label corrected loss $\widetilde{\ell}(h(x), \alpha) = \widetilde{g}_h(x, \alpha)$. To provide regret guarantees using SGD over bags in a parameter space $\mathcal{W}$ and loss function

$\ell\colon \mathcal{W} \times \mathcal{X} \times \mathcal{Y} \to \mathbb{R}$, we use EASYLLP to estimate the gradient of the loss function with respect to a parameter $\mathbf{w} \in \mathcal{W}$ by letting $g_{\mathbf{w}}(x, y) = \nabla_{\mathbf{w}} \ell(\mathbf{w}, x, y)$ and its corresponding soft label corrected function $\widetilde{g}_{\mathbf{w}}(x, \alpha) = \nabla \widetilde{\ell}(\mathbf{w}, x, \alpha)$.

# 5 ERM with Label Proportions

Given a hypothesis space $\mathcal{H}$, let $\ell$ be a loss function as defined in Section 2. Given a collection of bags $\mathcal{S} = \{(\mathcal{B}_i, \alpha_i), i \in [n]\}$ of size $k$, our learning algorithm simply finds $h \in \mathcal{H}$ that minimizes the empirical risk constructed via the soft label corrected loss from Eq. (3):

$$\sum_{i=1}^{n} \sum_{j=1}^{k} \widetilde{\ell}(h(x_{ij}), \alpha_i) \,. \tag{5}$$

The main advantage of our algorithm lies in its simplicity and generality. Indeed, our algorithm can be used for any loss function and any hypothesis set. This is in stark contrast, e.g., to the works of [22, 20], whose framework is only applicable to the logistic loss and (generalized) linear models. From a practical standpoint, our approach can also leverage existing learning infrastructures, as the only thing that needs to be specified is a different loss function — which in frameworks like Tensorflow, JAX and PyTorch requires only minimal coding. This differs from other approaches to assigning surrogate labels which may require solving combinatorial optimization problems (or relaxations thereof) like, e.g., [32, 8].

The following theorem provides learning guarantees for minimizing the above empirical loss. Our guarantees are given in terms of the well-known Rademacher complexity of a class of functions.

**Definition 5.1.** Let $\mathcal{Z}$ be an arbitrary input space and let $\mathcal{G} \subset \{g \colon \mathcal{Z} \to \mathbb{R}\}$ be a collection of functions over $\mathcal{Z}$. Let $\mathcal{D}$ be a distribution over $\mathcal{Z}$ and $\mathcal{S} = \{z_1, \dots, z_m\}$ be an i.i.d. sample. The Rademacher complexity of $G$ is given by $\mathfrak{R}_n(\mathcal{G}) = \frac{1}{n} \mathbb{E}_{\mathcal{S}, \boldsymbol{\sigma}} \left[ \sup_{g \in \mathcal{G}} \sum_{i=1}^{n} g(z_i) \sigma_i \right]$, where $\boldsymbol{\sigma} = (\sigma_1, \dots, \sigma_n) \in \{-1, 1\}^n$ is uniformly distributed.

**Theorem 5.2.** *Let $\delta > 0$, $\mathcal{S} = \{(\mathcal{B}_i, \alpha_i), i \in [n]\}$ be a collection of $n$ bags of size $k$.[4] Let $\sup_{\widehat{y}, y} \ell(\widehat{y}, y) \leq B$, and $C_n^k = 2 \left( \sqrt{2k \log(kn)} + 1 \right)$. Then the following bound holds uniformly for all $h \in \mathcal{H}$ with probability at least $1 - \delta$:*

$$\left| \mathcal{L}(h) - \frac{1}{nk} \sum_{i,j} \widetilde{\ell}(h(x_{ij}), \alpha_i) \right| \leq C_n^k \left( \mathfrak{R}_{kn}(\mathcal{H}_\ell^{(1)}) + \mathfrak{R}_{kn}(\mathcal{H}_\ell^{(0)}) \right) + \frac{4B}{n} + 4B \sqrt{\frac{k \log(2/\delta)}{2n}} \,,$$

*where $\mathcal{H}_\ell^{(1)} = \{x \to \ell(h(x), 1) \colon h \in \mathcal{H}\}$ and $\mathcal{H}_\ell^{(0)} = \{x \to \ell(h(x), 0) \colon h \in \mathcal{H}\}$.*

**Corollary 5.3.** *With the notation of the previous theorem, let $\widehat{h}$ denote the minimizer of (5). Then with probability at least $1 - \delta$ over the sampling process we have*

$$\mathcal{L}(\widehat{h}) \leq \min_{h \in \mathcal{H}} \mathcal{L}(h) + 2 \left( C_n^k \left( \mathfrak{R}_{kn}(H_\ell^{(1)}) + \mathfrak{R}_{kn}(H_\ell^{(0)}) \right) + \frac{4B}{n} + 4B \sqrt{\frac{k \log(2/\delta)}{2n}} \right) \,.$$

**Comparison to event level learning.** We can now compare the bound from Theorem 5.2 to standard learning bounds for instance level learning like that of [18]. Assuming we had access to a labeled i.i.d. sample $(x_{ij}, y_{ij})$ of size $kn$, Theorem 3.3 in [18] ensures that with probability at least $1 - \delta$ the following bounds holds for all $h \in \mathcal{H}$:

$$\left| \mathcal{L}(h) - \frac{1}{nk} \sum_{i,j} \ell(h(x_{ij}), y_{ij}) \right| \leq 2 \mathfrak{R}_{nk}(\mathcal{H}_\ell) + B \sqrt{\frac{\log(2/\delta)}{2nk}} \,, \tag{6}$$

where $\mathcal{H}_\ell = \{(x, y) \to \ell(h(x), y) \colon h \in \mathcal{H}\}$. Note that under the weak assumption that the Rademacher complexities $\mathfrak{R}_{kn}(\mathcal{H}_\ell^{(r)})$, $r \in \{0, 1\}$, are of the same order as $\mathfrak{R}_{kn}(\mathcal{H}_\ell)$, the main

---

[4]Our results can easily be extended to the case where bags have different sizes, provided the items are still i.i.d. The equal size assumption is made here mainly for notational convenience. If the bag sizes are constant with the number of bags $n$, then a rate of the form $1/\sqrt{n}$ can still be obtained. On the contrary, if the bag sizes scale with $n$, the actual rate will depend on the interplay between the bag sizes and $n$ itself.

difference between the bound in Theorem 5.2 and (6) is simply an extra factor $C_n^k \in \tilde{O}(\sqrt{k})$ in the complexity term and a factor $k$ multiplying the confidence term. That is, we achieve similar guarantees to event level learning by increasing the sample size by a factor of roughly $k$.

It is worth stressing the difference in flavor of the consistency result contained in Theorem 5.2 above and those in [17, 16]. For instance, in [16] (even with $m = 2$ bags) the consistency limit has to be interpreted "as the bag size $n_{tr} = n_1 + n_2$ goes to infinity". In our case, the bag size $k$ has to remain constant, and it is the number of bags $n$ that goes to infinity. This further strengthens our claim that the stream of literature that those papers are representative of are by no means subsuming our results. Moreover, unlike our paper, all results in these previous works, as presented, make assumptions about the loss function (e.g., square loss or cross entropy for [16], margin-based losses for [17]). The fact that we make no assumptions allows us to apply our debiasing procedure to any function $g(x, y)$ of two variables, hence we can debias, e.g., also the *gradient* of a loss function, enabling the principled usage of stochastic gradient descent procedures with only label proportion information. This is illustrated in the next section.

## 6  SGD with Label Proportions

We now focus on understanding the effect of label proportions on another very popular learning algorithm: stochastic gradient descent (SGD).

Proposition 4.2 delivers an unbiased estimate of the gradient which can be naturally plugged into any SGD algorithm (e.g., [27]), and one would hope for an upper bound on the excess risk if the learning task at hand leads to a convex optimization problem. The difficulty is that, even if each gradient in a given bag is individually unbiased, the gradients are correlated since they depend on the label proportion computed on the bag. A simple way around it is to pick a single item uniformly at random from the bag to update the model parameters. This is a slight departure from what we considered for ERM, but it both makes our SGD analysis easier and does not affect asymptotic performance.

In order to devise an SGD algorithm that can handle bagged data, first note that the debiasing technique introduced in Proposition 4.2 is very general and it applies to any measurable function, including the gradient of any loss function. Let us denote the gradient of the loss on example $(x, y)$ by $g_{\mathbf{w}}(x, y)$ where $\mathbf{w}$ is the parameter vector of the model and let its soft-label corrected version be $\widehat{g}_{\mathbf{w}}$. We study a version of projected SGD that picks one example per bag and uses the soft-label corrected gradient estimates (pseudocode is given in in Algorithm 1 in the Appendix). The excess risk of this algorithm depends on the squared norm of the debiased gradients, which is at most $k$ times larger than the variance of the original instance-level gradient based on Theorem 4.3. This observation results in the following risk bound.

**Theorem 6.1.** *Suppose that $F(\mathbf{w}) = \mathbb{E}[\ell_{\mathbf{w}}(x, y)]$ is convex, $\mathbb{E}[\|g_{\mathbf{w}_t}(x, y)\|^2] \leq G^2$ for all $t \in [n]$, and $\sup_{\mathbf{w}, \mathbf{w}' \in \mathcal{W}} \|\mathbf{w} - \mathbf{w}'\| \leq D$. Consider Algorithm 1, run with step size $\eta_t = 1/\sqrt{k \cdot t}$. Then for any $n > 1$ we have*

$$\mathbb{E}[F(\mathbf{w}_n) - F(\mathbf{w}^*)] \leq \sqrt{k} \cdot \left(D^2 + 5G^2\right) \frac{2 + \log(n)}{\sqrt{n}}.$$

This theorem quantifies the deterioration in the excess of the risk in terms of bag size when the algorithm is applied with LLP data. Recall that the error of SGD with $kn$ individually labeled samples decreases like $O\left(1/\sqrt{kn}\right)$. Compared with the above bound, and similar to the ERM case, the regret bound increases by a factor of $k$.

The most predominant learning frameworks used today fit deep models to data using SGD-like methods. Even though these problems are typically non-convex, we remark that Proposition 4.2 guarantees that the expected gradient steps of Algorithm 1 are equal to the expected steps of SGD using an instance-level loss. Moreover, Theorem 4.3 guarantees that the variance of the gradient estimates used by Algorithm 1 is not much larger than that of the instance-level case. In particular, it follows that whenever SGD is an effective algorithm for learning from instance-level data, Algorithm 1 should also be effective at learning from LLP data, provided that we have enough data to overcome the increased variance. This is thoroughly demonstrated in the next section. Finally, the only required modifications to any gradient-based training infrastructure to use EASYLLP are to replace the original labels $y_{ik}$ by label proportions and to implement the soft-label corrected loss. Taken together, these properties of EASYLLP make it especially well suited to modern machine learning pipelines.

# 7  Experiments

In this section we empirically evaluate EASYLLP, PROPMATCH, and two baseline methods to characterize how their performance depends on the bag size for a range of different learning tasks and underlying learning models.

**LLP Methods.**  We evaluate EASYLLP, PROPMATCH, and two baselines.[5] The first baseline is MEANMAP [22], which is a method specialized for learning linear logistic regression models from label proportions. We also compare against the method described in Section 3.2 of [8], which we denote by DA. The DA method computes a loss for each bag by creating synthetic labels for the examples that optimize a combination of the model's loss on the synthetic labels, and a divergence between the label proportion of the synthetic labels and the bag's label proportion. The DA method has an additional hyperparemeter, $\alpha$, which controls the weights on these two objectives. Full details of our DA implementation are in Appendix A.3.

**Datasets.**  We carry out experiments on four (binary classification) datasets: Binarized versions of MNIST [13] and CIFAR-10 [12], as well as the Higgs [3] and UCI adult datasets [11]. The labels in MNIST are replaced by labels indicating whether the digit is even or odd, and the CIFAR-10 labels are replaced by whether the original class is an animal or a machine. The MNIST images are resized to have shape $32 \times 32 \times 3$ to match CIFAR-10. For both image datasets, we normalize the images to have $\ell_2$ norm equal to one. No preprocessing is performed on the Higgs dataset. Finally, on UCI Adult, we rescale numerical features to the range $[0, 1]$ and one-hot encode categorical features.

**Models.**  We evaluate each LLP method on every dataset using a linear model. Additionally, for the image datasets we also evaluate using a small ConvNet, a large ConvNet, and Resnet-11 [9]. On Higgs we also use the DNN proposed by [3], which has 4 hidden layers, each with 300 units. Finally, on UCI Adult, we also use a NN with one hidden layer. Since the MEANMAP baseline is only applicable to learning linear models, we do not include it in the neural network experiments. Full details of the architectures are provided in Appendix A.2.

**Experimental Setup.**  For each combination of LLP method, dataset, model, and bag size, we train the model 10 times with different seeds and report the average test accuracy. To generate the label proportion data, on each run we shuffle the training data, partition it into consecutive bags of the desired size, and replace the original labels with label proportions computed from the bags. To tune the learning rate for each method, we report the highest accuracy achieved for learning rates in $\{0.00001, 0.0001, 0.0005, 0.001, 0.005, 0.01, 0.05\}$. For DA, we tune the learning rate and $\alpha$ using a grid search with the same learning rates and $\alpha \in \{0.0, 0.0001, 0.001, 0.01, 0.1, 0.5\}$. Note that DA is sensitive to the choice of $\alpha$, and it is unclear how to tune $\alpha$ without access to event-level data. In all cases we use the Adam [10] optimizer, binary crossentropy loss, minibatches of size 512, and 20 training passes through the data. Finally, for the two image datasets, we decay the learning rate after 40%, 60%, 80%, and 90% of the training passes by factors 10, 100, 1000, and 5000, respectively.

**Results.**  Figure 2 depicts the accuracy achieved by each method on a selection of datasets and models for a range of bag sizes. The selected dataset and model combinations are representative of the full set of experimental results, which are included in Appendix A.5. Across all datasets, either EASYLLP or PROPMATCH achieves the highest test accuracy for large bag sizes. In agreement with our theory, the performance of EASYLLP decreases predictably with the bag size (due to the increased variance of gradient estimates). On the other hand, we find that DA often performs very well for small bag sizes, but accuracy drops rapidly at a dataset-dependent threshold. PROPMATCH is a competitive baseline, specifically when the underlying learning model is well specified for the task, which is in line with our finding in Section 3. For instance, when using the Resnet-11 model, PROPMATCH has a higher performance than EASYLLP on both MNIST and CIFAR-10. A partial explanation is due to Theorem 4.3. Indeed, it is well known that models trained under logistic regression become overconfident on their predictions. However, our theory suggests that the variance of the gradient increases proportional to $\|g(x,0) - g(x,1)\|$ which rapidly becomes unbounded as

---

[5]Due to the differences in problem setup between our work and that of other papers, like [17, 16], it is unclear how a fair experimental comparison to those papers should be performed. For this very reason, we have deliberately avoided to experimentally compare to the algorithms in those papers.

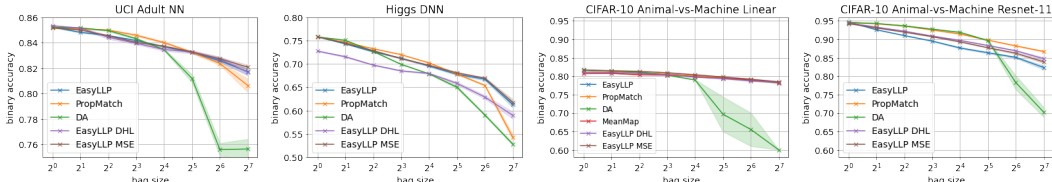

Figure 2: Average test accuracy for each LLP method on selected datasets and models. Error bars show one standard error in the mean.

the model becomes more confident. To test this hypothesis, we trained a model using a double-sided hinge loss, labeled "EASYLLP DHL" (details in Appendix A). This loss induces a small variance term in Theorem 4.3 and indeed we see that this partially closes the gap to PROPMATCH. On the other hand, PROPMATCH's performance degrades more rapidly than EASYLLP's as the bag size increases in the presence of model mis-specification or higher noise in the data (first three plots in Figure 2).

**Loss tracking.**   Another strength of EASYLLP is that the loss estimates computed during training are unbiased estimates of the true training loss. In comparison, the losses minimized by all other methods we consider do not have clear connections to the true training loss and it is unclear how to monitor training performance. In Appendix A.6 we demonstrate this empirically.

## 8    Conclusions and Limitations

We have studied the problem of learning from label proportions in the case where bags are drawn i.i.d. We have introduced EASYLLP, a novel and flexible approach to LLP for classification that is widely applicable and has well-developed theory. In particular, we have shown how to use EASYLLP to estimate the expected value of *any* function of $(x, y)$ pairs from labeled data, and applied these results to proving ERM sample complexity guarantees and convergence guarantees for SGD. In both cases, we have shown that the LLP performance of EASYLLP is only a factor $k$ worse than when learning from event-level data. We have also elucidated important theoretical properties of the folklore PROPMATCH algorithm which are suggestive of the practical scenarios where it is advisable to make use of it. Finally, we have carried out an extensive empirical evaluation of EASYLLP, PROPMATCH, and two baseline methods on diverse tasks and learning models, and identified relative performance trends.

The results in this work are limited to the special case of LLP where the examples in each bag are drawn i.i.d. from an underlying distribution. While this is a common and important special case, in future work we would like to extend our results to handle cases where the bags need not be i.i.d. Similarly, we hope to study the theoretical properties of EASYLLP more fully in the multiclass case, and to have a complete characterization of when PROPMATCH is consistent. We also believe that it may be possible to further decrease the variance of EASYLLP estimates by replacing the label marginal $p$ in (3) with bag-specific predictions. Finally, while the LLP framework provides an intuitive form of privacy protection, we would like to explore connections with differential privacy where, for example, each bag's label proportion is computed by a differentially private mechanism.

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
