# A  Experiment Details and Complete Results

## A.1  Computing Environment.

All experiments were carried out concurrently on a cluster of NVIDIA Tesla p100 GPUs using Tensorflow and Keras.

## A.2  Model Architectures

In this section we describe in detail each of the model architectures we use in our experiments.

**Small ConvNet.**  Our small ConvNet consists of the following layers:

- A convolutional layer with 32 kernels of size $3 \times 3$ and ReLU activation.
- A max pooling layer with pool $2 \times 2$.
- A convolutional layer with 64 kernels of size $3 \times 3$ and ReLU activation.
- A max pooling layer with pool size $2 \times 2$.
- A flatten layer.
- A dropout layer with drop rate 0.5.
- A dense output layer with 1 unit and sigmoid activation.

For inputs of shape $32 \times 32 \times 3$, this model has 21,697 parameters.

**Large ConvNet.**  Our large ConvNet model consists of the following layers:

- A convolutional layer with 32 kernels of size $3 \times 3$, padding, and ReLU activation.
- A convolutional layer with 32 kernels of size $3 \times 3$ with ReLU activation.
- A max pooling layer with pool size $2 \times 2$.
- A dropout layer with drop rate 0.25.
- A convolutional layer with 64 kernels of size $3 \times 3$, padding, and ReLU activation.
- A convolutional layer with 64 kernels of size $3 \times 3$ and ReLU activation.
- A max pooling layer with pool size $2 \times 2$.
- A dropout layer with drop rate 0.25.
- A flatten layer.
- A dense layer with 512 units and ReLU activation.
- A dropout layer with drop rate 0.5.
- A dense output layer with 1 unit and sigmoid activation.

For inputs of size $32 \times 32 \times 3$, this model has 1,246,241 parameters.

**Resnet-11.**  We use the standard Resnet-11 [9] architecture which, for inputs of size $32 \times 32 \times 3$ and binary output has 296,865 parameters.

**Higgs DNN.**  Following Baldi et al. [3], we use a DNN consisting of the following layers:

- A dense layer with 300 units and ReLU activation.
- A dense layer with 300 units and ReLU activation.
- A dense layer with 300 units and ReLU activation.
- A dense layer with 300 units and ReLU activation.
- A dense layer with 1 unit and sigmoid activation.

This model has 279,901 parameters. Note that Baldi et al. [3] use *tanh* activations on the hidden layers and a small amount of regularization, but we did not find this to improve instance-level performance in our experiments.

**UCI Adult NN.** For the UCI Adult dataset, we use a neural network with the following layers:

- A dense layer with 32 units and ReLU activation.
- A dense layer with 1 unit and sigmoid activation.

This model has 3,521 parameters.

### A.3 DA Details

Section 3.2 of [8] defines a combinatorial loss that attempts to guess the true labels of each example in a bag based on the current model. Translating into our notation and specializing to the binary case, for a bag $\mathcal{B} = \{x_1, \ldots, x_k\}$ with label proportion $\alpha$, the loss of predictions $\hat{y}_1, \ldots, \hat{y}_k \in [0, 1]$ is given by

$$\ell^{\mathrm{comb}}(\hat{y}_1, \ldots, \hat{y}_k, \alpha) = \min_{t_1, \ldots, t_k \in \{0,1\}} \frac{\alpha}{n} \sum_{i=1}^{k} \ell(\hat{y}_i, t_i) + (1 - \alpha) \operatorname{div}\left(\frac{1}{k} \sum_{i=1}^{k} t_i, \alpha\right),$$

where[6] $\ell$ is the original loss, $t_1, \ldots, t_k$ are "guessed labels", and $\operatorname{div}$ is a notion of divergence between two distributions (e.g., the KL divergence).

The main method proposed by Dulac-Arnold et al. [8] is to optimize a convex relaxation of $\ell^{\mathrm{comb}}$ because the multiclass version of the above objective is computationally expensive to evaluate. However, they observe that in the special case of binary classification, the optimization can be solved exactly by sorting the data in increasing order according to $\ell(y_i, 1) - \ell(y_i, 0)$. After sorting, the optimal solution over the guessed labels $t_1, \ldots, t_k$ always sets $t_i = 1$ for a prefix of the sorted list. In particular, the optimum can be found in linear time by trying all possible prefix lengths. The overall running time of this approach is $O(k \log k)$ per bag.

We implement the sorting-based version of the combinatorial loss and set $\operatorname{div}(p, q)$ to be the symmetrized KL-divergence between Bernoulli distributions with parameters $p$ and $q$. As discussed in the main body, in each application of DA we tune both the learning rate and the value for the $\alpha$ parameter using a grid search.

### A.4 EASYLLP Double Hinge Loss

As discussed in the main body, the variance computed in Theorem 4.3 becomes large when the loss is binary cross-entropy and the model makes confident predictions (close to 0 or 1). Intuitively, this is because the soft-label corrected loss estimate is a weighted combination of the loss if the true label had been zero and if the true label had been one. For the cross-entropy loss, as the model becomes more confident, one of these two terms diverges (both when estimating the loss and its gradient). In this section we describe an alternative loss function that achieves high accuracy with EASYLLP in some cases where the binary cross-entropy performs poorly.

Let $\hat{z}$ be the logit output by a model for an example with label $y$. We define the double hinge loss as

$$\ell_{\mathrm{DHL}}(\hat{z}, y) = \max(0, 1 - m, m - 2), \qquad \text{where} \qquad m = \hat{z}(2y - 1) = \begin{cases} \hat{z} & \text{if } y = 1 \\ -\hat{z} & \text{if } y = 0. \end{cases}$$

Figure 3 compares $\ell_{\mathrm{DHL}}$ to the binary cross-entropy as a function of the margin $m$.

This loss function is designed so that that for any predicted logit $\hat{z}$, the scale of $\ell_{\mathrm{DHL}}(\hat{z}, 0)$ and $\ell_{\mathrm{DHL}}(\hat{z}, 1)$ is roughly the same (and so are the derivatives w.r.t. $\hat{z}$). Recall that the soft-label corrected loss can be written as $w_0 \cdot \ell_{\mathrm{DHL}}(\hat{z}, 0) + w_1 \cdot \ell_{\mathrm{DHL}}(\hat{z}, 1)$ where $w_0$ and $w_1$ are (possibly negative) weights that sum to one. When $\ell_{\mathrm{DHL}}(\hat{z}, 0)$ and $\ell_{\mathrm{DHL}}(\hat{z}, 1)$ have comparable scales, this weighted sum also has a comparable scale.

In Appendix A.5 we plot the performance of EASYLLP for every task using the double hinge loss. For CIFAR-10 Animal-vs-Machine and MNIST Even-vs-Odd, EASYLLP using the double hinge loss has much closer performance to PROPMATCH (the best performing method) than EASYLLP using the binary cross-entropy loss. The only learning task where the double hinge loss performs significantly worse than the binary cross-entropy is UCI Adult with a linear model.

---

[6] Dulac-Arnold et al. [8] only consider the binary cross-entropy loss, but the generalization to other losses is straightforward.

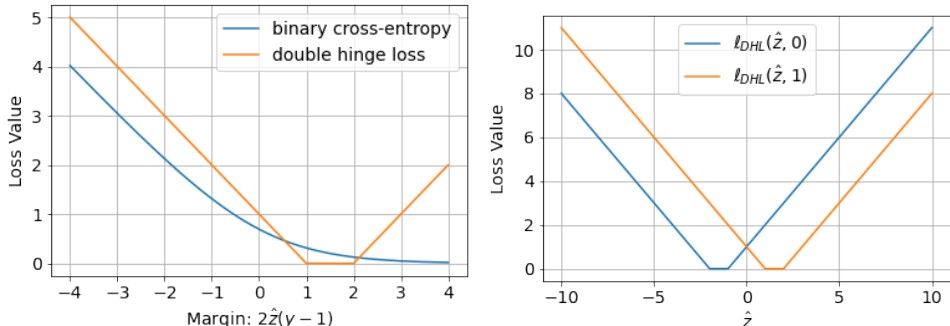

Figure 3: Left: Comparison between the double hinge loss and the binary cross-entropy as functions of the margin $2\hat{z}(y - 1)$. Right: The components $\ell_{\mathrm{DHL}}(\hat{z}, 0)$ and $\ell_{\mathrm{DHL}}(\hat{z}, 1)$. For all values of $\hat{z}$, the magnitude of the loss (and the derivative w.r.t. $\hat{z}$) are comparable.

### A.5 Complete Experimental Results

In this section we present our empirical results for all combinations of LLP method, dataset, model, and bag size. Figure 4 shows the results for the CIFAR-10 Animal-vs-Machine task. Figure 5 shows the results for the MNIST Even-vs-Odd task. Figure 6 shows the results for the Higgs dataset. Finally, Figure 7 shows the results for the UCI Adult dataset.

For all datasets, we include two variants of EASYLLP. The first, labeled EASYLLP MSE minimizes the mean squared error on the training data (rather than the binary cross-entropy). Similarly, the second, labeled EASYLLP DHL minimizes the double hinge loss defined in Appendix A.4.

In addition to our remarks in Section 7, we observe that the DA method has relatively low performance compared to other methods particularly when we train models with low complexity. In particular, when training linear models on every dataset, the accuracy of DA drops much earlier than the other methods. The combinatorial loss optimized by DA reinforces the current model's predictions. A possible explanation for the relatively poor performance with simple models is that, if the model is incapable of making accurate predictions, the loss will often be reinforcing incorrect labels. More broadly, this issue should arise for any DA-like method that attempts to use the model's current predictions to guess the labels of examples within a bag.

### A.6 Loss tracking

In this section, we show that the EASYLLP soft-label corrected loss closely tracks the model's actual training loss, while the losses optimized by PROPMATCH and DA do not. This close agreement between the EASYLLP loss estimates and the true training loss makes transferring a practitioner's domain expertise from the non-LLP version of a learning problem easier.

We train the Higgs DNN model with a bag size of 32 using the optimal parameters from our grid search for EASYLLP, PROPMATCH, and DA. For each minibatch, we compute the value of the loss being minimized by each method on that minibatch, as well as the actual training loss of the model using the original (non-proportion) labels. Figure 8 depicts sliding window averages of the loss each method optimizes together with the actual training loss. The window size is 2000 minibatches, so each point on the curve is the average of the loss over the previous 2000 minibatches.

We find that there is close agreement between the EASYLLP loss estimates and the actual loss. The proportion matching loss decreases as training proceeds, but at a much slower rate than the actual loss. Finally, for DA, the value of the combinatorial loss is very close to zero throughout training, likely due to the fact that the loss constructs synthetic labels that agree with the model predictions.

## B  Proofs

Throughout this appendix, we denote by $\mathbb{1}_A$ the indicator function of the predicate $A$ at argument.

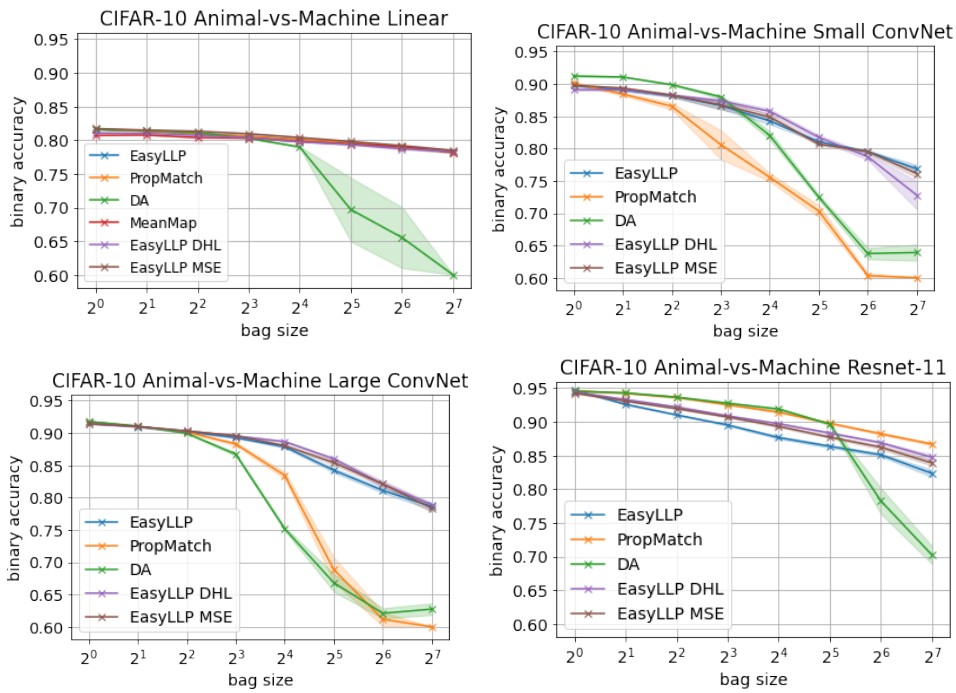

Figure 4: Experimental results for CIFAR-10 Animal-vs-Machine with various models.

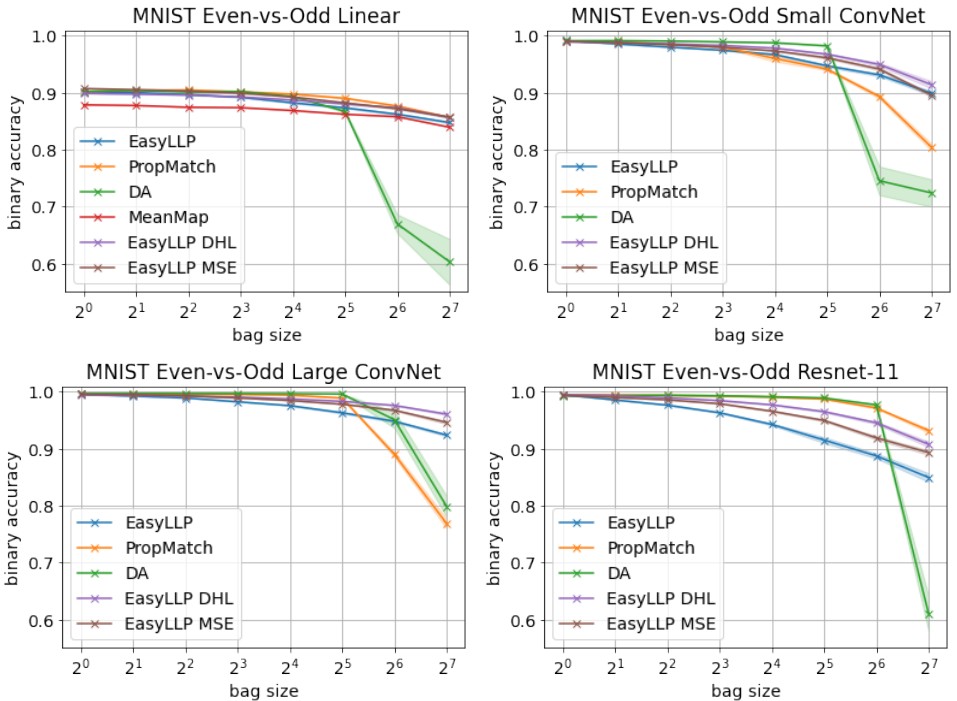

Figure 5: Experimental results for MNIST Even-vs-Odd with various models.

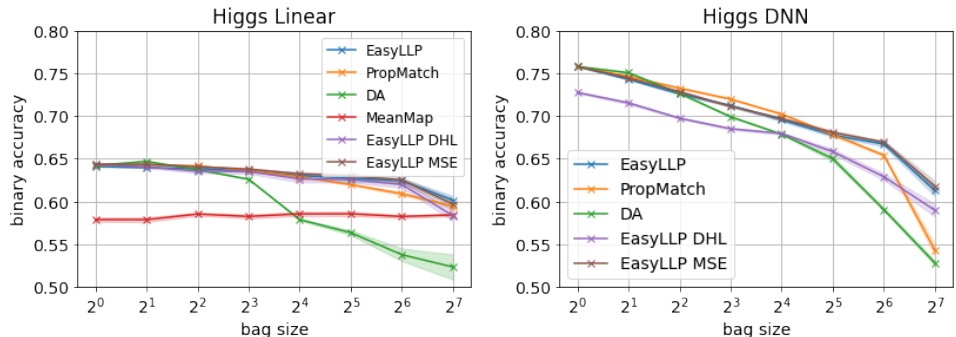

Figure 6: Experimental results for Higgs with various models.

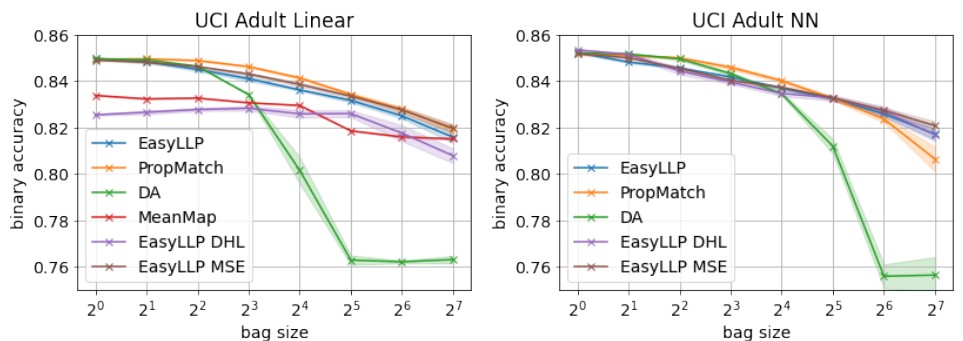

Figure 7: Experimental results for UCI Adult with various models.

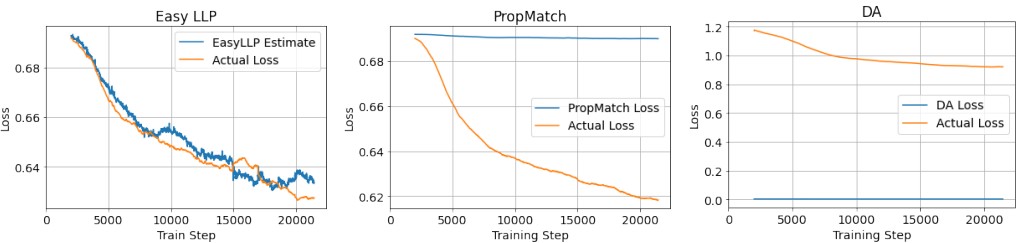

Figure 8: Comparison of the loss values optimized by the EASYLLP, PROPMATCH, and DA methods and the actual training loss of the model (evaluated from event-level access to the data). Results are averaged over a sliding window of 2000 steps.

## B.1 Missing proofs from Section 3

**Theorem 3.2.** *Assume $\mathcal{H}$ is such that the function $h^*: x \mapsto \mathbb{P}(y = 1|x)$ is in $\mathcal{H}$. Let $\ell : \mathbb{R} \times \mathbb{R} \to \mathbb{R}^+$ be such that, for any random variable $Z$, $q = \mathbb{E}[Z]$ is the* unique *solution of $\min_{r \in \mathbb{R}} \mathbb{E}_Z[\ell(r, Z)]$. Then, $h^*$ is a minimizer of* (2). *Moreover, every other minimizer $h$ satisfies $\mathbb{P}(h(x){=}h^*(x)) = 1$.*

*Proof.* We begin by showing that $h^*$ is minimizer of the objective. To do so, we lower bound the objective in (2) as follows:

$$\min_{h \in \mathcal{H}} \mathbb{E}_{\mathcal{B}, \alpha} \left[ \ell \left( \frac{1}{k} \sum_{x \in \mathcal{B}} h(x), \alpha \right) \right] \geq \mathbb{E}_{\mathcal{B}} \left[ \min_{h \in \mathcal{H}} \mathbb{E}_{\alpha} \left[ \ell \left( \frac{1}{k} \sum_{x \in \mathcal{B}} h(x), \alpha \right) \middle| \mathcal{B} \right] \right]$$

$$\geq \mathbb{E}_{\mathcal{B}} \left[ \min_{r \in \mathbb{R}} \mathbb{E}_{\alpha} \left[ \ell \left( r, \alpha \right) \middle| \mathcal{B} \right] \right],$$

The inequality follows from the fact that the image of the the the mapping $h \mapsto \frac{1}{k} \sum_{x \in \mathcal{B}} h(x)$ is only a subset of $\mathbb{R}$. We now use the property of the loss function to see that the last expression is equivalent to

$$\mathbb{E}_{\mathcal{B}}\left[\mathbb{E}_{\alpha}\left[\ell\left(\mathbb{E}[\alpha|\mathcal{B}], \alpha\right)|\,\mathcal{B}\right]\right] = \mathbb{E}_{\mathcal{B}, \alpha}\left[\ell\left(\mathbb{E}[\alpha|\mathcal{B}], \alpha\right)\right] \;.$$

But since $\alpha$ is the average label of examples in the bag it follows by linearity of conditional expectation that $\mathbb{E}[\alpha|\mathcal{B}] = \frac{1}{k} \sum_{x \in \mathcal{B}} \mathbb{P}(y = 1|x) = \frac{1}{k} \sum_{x \in \mathcal{B}} h^*(x)$. Putting everything together we see that

$$\min_{h \in \mathcal{H}} \mathbb{E}_{\mathcal{B}, \alpha}\left[\ell\left(\frac{1}{k} \sum_{x \in \mathcal{B}} h(x), \alpha\right)\right] \geq \mathbb{E}\left[\ell\left(\frac{1}{k} \sum_{x \in \mathcal{B}} h^*(x), \alpha\right)\right].$$

Since $h^* \in \mathcal{H}$ it follows that $h^*$ is a minimizer of the proportion matching loss. We proceed to demonstrate that it is the unique minimzer. Let $h_0$ be a hypothesis such that $\mathbb{P}(h_0(x) \neq h^*(x)) > 0$. Note that since $\mathbb{E}[\alpha|\mathcal{B}] = \frac{1}{k} \sum_{x \in B} h^*(x)$ is the unique solution of $\min_{r \in \mathbb{R}} \mathbb{E}_{\alpha}\left[\ell\left(r, \alpha\right)|\mathcal{B}\right]$, if we prove that $\mathbb{P}_{\mathcal{B}}(\frac{1}{k} \sum_{x \in \mathcal{B}} h_0(x) \neq \frac{1}{k} \sum_{x \in \mathcal{B}} h^*(x)) > 0$, then $h_0$ cannot be a minimizer of the proportion matching loss. This follows from the following simple observation:

$$\mathbb{P}_{\mathcal{B}}\left(\frac{1}{k} \sum_{x \in \mathcal{B}} h_0(x) \neq \frac{1}{k} \sum_{x \in \mathcal{B}} h^*(x)\right) = \mathbb{P}_{\mathcal{B}}\left(\frac{1}{k} \sum_{x \in \mathcal{B}} h_0(x) > \frac{1}{k} \sum_{x \in \mathcal{B}} h^*(x)\right) + \mathbb{P}_{\mathcal{B}}\left(\frac{1}{k} \sum_{x \in \mathcal{B}} h_0(x) < \frac{1}{k} \sum_{x \in \mathcal{B}} h^*(x)\right)$$

$$\geq \mathbb{P}(h_0(x) > h^*(x) \; \forall x \in \mathcal{B}) + \mathbb{P}(h_0(x) < h^*(x) \; \forall x \in \mathcal{B})$$

$$= \mathbb{P}(h_0(x) > h^*(x))^k + \mathbb{P}(h_0(x) < h^*(x))^k$$

$$> 0 \;,$$

where we used the independence of examples in the bag for the last equality, and the fact that

$$\mathbb{P}(h_0(x) > h^*(x)) + \mathbb{P}(h_0(x) < h^*(x)) = \mathbb{P}(h_0(x) \neq h^*(x)) > 0$$

for the last inequality. $\qquad \square$

**Corollary 3.3.** *Let $\mathcal{H}$ satisfy the conditions of Theorem 3.2. Then $h^*$ as defined above is the unique minimizer for the proportion matching loss when $\ell$ is the square loss or the binary cross-entropy loss.*

*Proof.* Let $Z$ be an arbitrary random variable and let $\ell(p, Z) = (p - Z)^2$. It is a well known fact that $\mathbb{E}[Z]$ is the unique minimizer $\mathbb{E}[\ell(p, Z)]$.

We now focus on the binary cross entropy loss. Fix $k > 0$ and consider a random variable $Z$ taking values on the set $\{i/k \colon i = 0, \ldots, k\}$. Let $\ell$ now denote the cross entropy loss given by:
$$\ell(p, Z) = -Z \log p - (1 - Z) \log(1 - p)$$
Then $\mathbb{E}[\ell(p, Z)] = -\mathbb{E}[Z] \log p - (1 - \mathbb{E}[Z]) \log(1 - p)$. Taking the derivative with respect to $p$ and setting it to zero, shows that the unique minimizer is given by $p = \mathbb{E}[Z]$. $\qquad \square$

## B.2 Proof of Proposition 4.2

**Proposition 4.2.** *Given a sample $(x_1, y_1), \ldots, (x_k, y_k)$ drawn i.i.d. according to $\mathcal{D}$, let $(\mathcal{B}, \alpha)$ be the corresponding labeled bag of size $k$, for some $k \geq 1$. Let $g \colon \mathcal{X} \times \mathcal{Y} \to \mathbb{R}^d$ be any (measurable) function, for some output dimension $d \geq 1$, and $\widetilde{g} \colon \mathcal{X} \times [0, 1] \to \mathbb{R}^d$ be its associated soft-label corrected function. Then for every element $x_j \in \mathcal{B}$ we have*
$$\mathbb{E}_{(\mathcal{B}, \alpha)}[\widetilde{g}(x_j, \alpha)] = \mathbb{E}_{(x, y) \sim \mathcal{D}}[g(x, y)] \;.$$

*Proof.* Let $(x_1, y_1), \ldots, (x_k, y_k)$ be a sample drawn i.i.d. from $\mathcal{D}$ and let $\alpha = \frac{1}{k} \sum_{i=1}^{k} y_i$ be the label proportion. Fix $j$ and let $S_j = \sum_{i \,:\, i \neq j} y_i$. Note that $\widetilde{g}(x_j, \alpha)$ can be rewritten as:
$$\widetilde{g}(x_j, \alpha) = (\mathbb{1}_{y_j=1} + S_j - (k-1)p)g(x_j, 1) + (\mathbb{1}_{y_j=0} + ((k-1)p - S_j))g(x_j, 0)$$
$$= g(x_j, y_j) + (S_j - \mu_p)g(x_j, 1) + (\mu_p - S_i)g(x_j, 0)$$
$$= g(x_j, y_j) + (S_j - \mu_p)(g(x_j, 1) - g(x_j, 0)) \;,$$
where $\mu_p = (k - 1)p$. We therefore have the following slightly stronger statement about the unbiasedness of $\widetilde{g}$:
$$\mathbb{E}[\widetilde{g}(x_j, \alpha)|x_j, y_j] = g(x_j, y_j)$$
Taking an outer expectation w.r.t. $(x_j, y_j)$ concludes the proof. $\qquad \square$

## B.3 Proof of Theorem 4.3

In this section we prove Theorem 4.3, which we recall now:

**Theorem 4.3.** *Let $g\colon \mathcal{X} \times \mathcal{Y} \to \mathbb{R}^d$ be such that $\sup_{x,y} \|g(x,y)\|^2 \leq M$, and denote by $\widetilde{g}$ its associated soft labeled corrected function. Also, set for brevity $g_0 = g(x,0)$ and $g_1 = g(x,1)$ and, for each $j \in [k]$, $\widetilde{g}_j = \widetilde{g}(x_j, \alpha)$. Then, for any size $k \geq 1$ and any $j \in [k]$,*

$$\mathbb{E}[\|\widetilde{g}_j\|^2] = \mathbb{E}[\|g(x_j, y_j)\|^2] + (k-1)p(1-p)\,\mathbb{E}\left[\|g_0 - g_1\|^2\right]$$

$$\mathbb{E}\left[\left\|\frac{1}{k}\sum_{i=1}^{k}\widetilde{g}_i\right\|^2\right] \leq \mathbb{E}[\|\widetilde{g}_j\|^2] . \tag{4}$$

*Moreover, there exists a universal constant $C$ such that*

$$\mathbb{E}\left[\left\|\frac{1}{k}\sum_{i=1}^{k}\widetilde{g}_i\right\|^2\right] \leq C + kp(1-p)\left\|\mathbb{E}[g_0 - g_1]\right\|^2 ,$$

*where $p = \mathbb{P}_{(x,y)\sim\mathcal{D}}(y = 1)$.*

For simplicity of notation let us define the following quantities:

$$\widetilde{g}_i = \widetilde{g}(x_i, \alpha) \qquad g_{i0} = g(x_i, 0), \qquad g_{i1} = g(x_i, 1) .$$

Let also $A = k(\alpha - p) + p$. With this notation we have that

$$\widetilde{g}_i = Ag_{i1} + (1 - A)g_{i0}.$$

The proof of Theorem 4.3 will be a consequence of the following lemmas.

**Lemma B.1.** *Using the above notation, the following inequality holds*

$$\mathbb{E}\left[\left\|\frac{1}{k}\sum_{i=1}^{k}\widetilde{g}_i\right\|^2\right] \leq \mathbb{E}[\|\widetilde{g}_1\|^2] .$$

*Proof.* By simple linear algebra and the fact that the $\widetilde{g}_i$ are equally distributed we have

$$\mathbb{E}\left[\left\|\frac{1}{k}\sum_{i=1}^{k}\widetilde{g}_i\right\|^2\right] = \frac{1}{k}\,\mathbb{E}[\|\widetilde{g}_1\|^2] + \frac{k-1}{k}\,\mathbb{E}[\langle\widetilde{g}_1, \widetilde{g}_2\rangle] . \tag{7}$$

Moreover,

$$0 \leq \mathbb{E}[\|\widetilde{g}_1 - \widetilde{g}_2\|^2] = 2\,\mathbb{E}[\|\widetilde{g}_1\|^2] - 2\,\mathbb{E}[\langle\widetilde{g}_1, \widetilde{g}_2\rangle]$$

implies

$$\mathbb{E}[\langle\widetilde{g}_1, \widetilde{g}_2\rangle] \leq \mathbb{E}[\|\widetilde{g}_1\|^2] .$$

Replacing this inequality in (7) yields

$$\mathbb{E}\left[\left\|\frac{1}{k}\sum_{i=1}^{k}\widetilde{g}_i\right\|^2\right] \leq \mathbb{E}[\|\widetilde{g}_1\|^2] ,$$

which is the claimed result. $\qquad\square$

**Lemma B.2.** *Using the previous notation, for every $i \leq k$ we have*

$$\mathbb{E}[\|\widetilde{g}_i\|^2] = \mathbb{E}[\|g(x_i, y_i)\|^2] + (k-1)p(1-p)\,\mathbb{E}[\|g(x,1) - g(x,0)\|^2] \tag{8}$$

*Proof.* Fix an index $i$ and rewrite $A$ as

$$A = \sum_{j=1}^{k} y_j - (k-1)p = y_i + \sum_{j\neq i} y_j - (k-1)p = y_i + B$$

where $B$ is a centered binomial random variable of parameters $(k-1, p)$ independent of $x_i, y_i$. This entails,
$$\mathbb{E}[B] = 0 \quad \text{and} \quad \mathbb{E}[B^2] = (k-1)p(1-p) . \tag{9}$$
Setting for brevity $g_{i1} = g(x_i, 1)$, $g_{i0} = g(x_i, 0)$, we thus have
$$\widetilde{g}_i = (B + y_i)g_{i1} + (1 - B - y_i)g_{i0} = y_i(g_{i1} - g_{i0}) + g_{i0} + B(g_{i1} - g_{i0})$$
Also note that if $y_i = 1$, then $y_i(g_{i1} - g_{i0}) + g_{i0} = g_1$ and if $y_i = 0$ the previous expression is equal to $g_0$. Thus we have that $y_i(g_{i1} - g_{i0}) + g_{i0} = g(x_i, y_i)$ and
$$\begin{aligned}
\mathbb{E}[\|\widetilde{g}_i\|^2] &= \mathbb{E}[\|g(x_i, y_i) + B(g_{i1} - g_{i0})\|^2] \\
&= \mathbb{E}[\|g(x_i, y_i)\|^2] + 2\,\mathbb{E}[B\langle g(x_i, y_i), (g_{i1} - g_{i0})\rangle] + \mathbb{E}[B^2\|g_{i1} - g_{i0}\|^2] \\
&= \mathbb{E}[\|g(x_i, y_i)\|^2] + 2\,\mathbb{E}[B]\,\mathbb{E}[\langle g(x_i, y_i), (g_{i1} - g_{i0})\rangle] + \mathbb{E}[B^2]\,\mathbb{E}[\|g_{i1} - g_{i0}\|^2],
\end{aligned}$$
where we have used the fact that $g(x_i, y_i), g_{i0}, g_{i1}$ are all functions of $x_i, y_i$ and $B$ is independent of of these variables. Using (9) to replace the terms depending on $B$ we obtain the result. $\square$

**Lemma B.3.** *Let $M = \sup_{x,y} \|g(x, y)\|$. For any pair of indices $i, j$ the following inequality holds*
$$\mathbb{E}[\langle \widetilde{g}_i, \widetilde{g}_j \rangle] \le (k-2)p(1-p)\|\,\mathbb{E}[g_{i1} - g_{i0}]\|^2 + 36M$$

*Proof.* Fix $i, j$. As in the previous lemma, let us rewrite $A$ as
$$A = y_i + y_j - p + \sum_{r \ne i,j} y_r - (k-2)p \tag{10}$$
$$= R + B \tag{11}$$
where $R = y_i + y_j - p$ and $B$ is a centered binomial random variable of parameters $(k-2, p)$. Moreover $B$ is independent of $(x_i, x_j, y_i, y_j)$. We proceed to calculate the desired expectation

$$\begin{aligned}
\mathbb{E}[\langle \widetilde{g}_i, \widetilde{g}_j \rangle] &= \mathbb{E}[\langle A(g_{i1} - g_{i0}) + g_{i0}, A(g_{j1} - g_{j0}) + g_{j0} \rangle] \\
&= \mathbb{E}[\langle B(g_{i1} - g_{i0}) + R(g_{i1} - g_{i0}) + g_{i0}, B(g_{j1} - g_{j0}) + R((g_{j1} - g_{j0}) + g_{j0} \rangle] ,
\end{aligned}$$

where, again, $g_{i1} = g(x_i, 1)$, and $g_{i0} = g(x_i, 0)$. Using a similar argument as in the previous lemma, it is not hard to see that the above expression simplifies to:
$$\mathbb{E}[\langle \widetilde{g}_i, \widetilde{g}_j \rangle] = (k-2)p(1-p)\,\mathbb{E}[\langle g_{i1} - g_{i0}, g_{j1} - g_{j0} \rangle] + \mathbb{E}[\langle R((g_{i1} - g_{i0}) + g_{i0}, R((g_{j1} - g_{j0}) + g_{j0} \rangle] .$$
Using the fact that $g_{i1} - g_{i0}$ and $g_{j1} - g_{g0}$ are independent and identically distributed as well as Cauchy-Schwartz inequality for the second term we obtain the following bound
$$\mathbb{E}[\langle \widetilde{g}_i, \widetilde{g}_j \rangle] \le (k-2)p(1-p)\|\,\mathbb{E}[g_{i1} - g_{i0}]\|^2 + \mathbb{E}[9MR^2] \le (k-2)p(1-p)\|\,\mathbb{E}[g_{i1} - g_{i0}]\|^2 + 36M ,$$
as claimed. $\square$

**Lemma B.4.** *Under the same notation as the previous lemma we have*
$$\mathbb{E}\left[\left\|\frac{1}{k}\sum_{i=1}^{k} \widetilde{g}_i\right\|^2\right] \le \frac{1}{k}\left(\mathbb{E}[\|g(x_i, y_i)\|^2] + (k-1)p(1-p)\,\mathbb{E}[\|g(x, 1) - g(x, 0)\|\|^2]\right)$$
$$+ \frac{(k-1)}{k}\left(36M^2 + (k-2)p(1-p)\|\,\mathbb{E}[g(x, 1) - g(x, 0)]\|^2\right)$$

*Proof of Theorem 4.3.* Using (7) we can write
$$\mathbb{E}\left[\left\|\frac{1}{k}\sum_{i=1}^{k} \widetilde{g}_i\right\|^2\right] = \frac{1}{k}\mathbb{E}[\|\widetilde{g}_1\|^2] + \frac{k-1}{k}\,\mathbb{E}[\langle \widetilde{g}_1, \widetilde{g}_2 \rangle] .$$
By applying Lemma B.2 and Lemma B.3 we can bound the above expression as
$$\mathbb{E}\left[\left\|\frac{1}{k}\sum_{i=1}^{k} \widetilde{g}_i\right\|^2\right] \le \frac{1}{k}\left(\mathbb{E}[\|g(x_i, y_i)\|^2] + (k-1)p(1-p)\,\mathbb{E}[\|g(x, 1) - g(x, 0)\|\|^2]\right)$$
$$+ \frac{(k-1)}{k}\left(36M^2 + (k-2)p(1-p)\|\,\mathbb{E}[g(x, 1) - g(x, 0)]\|^2\right) .$$
We let constant $C$ in the statement of the theorem depend on bound $M$, and the proof is concluded. $\square$

## B.4  Proof of Theorem 5.2

The proof of the theorem depends on the following proposition.

**Proposition B.5.** *For any $\eta > 0$ define $c(k, \eta, p) = \sqrt{\frac{\log(1/\eta)}{2k}}$. Let*

$$\widetilde{\ell}_\eta(\widehat{y}, \alpha) = \widetilde{\ell}(\widehat{y}, \alpha)\mathbb{1}_{|\alpha - p| \leq c(k, \eta)}.$$

*be a truncation of $\widetilde{\ell}$. Let $(\mathcal{B}, \alpha)$ be a labeled bag. Let $x \in \mathcal{B}$. With probability at least $1 - \eta$ over the choice of $(\mathcal{B}, \alpha)$, we have $\widetilde{\ell}_\eta(h(x), \alpha) = \widetilde{\ell}(h(x), \alpha)$ for all $h \in \mathcal{H}$.*

*Proof.* For any $h$, note that both losses agree unless $(\alpha - k) \geq c(k, \eta)$ but by Hoeffding's inequality this occurs with probability at most $\eta$. □

We now proceed to prove Theorem 5.2.

**Theorem 5.2.** *Let $\delta > 0$, $\mathcal{S} = \{(\mathcal{B}_i, \alpha_i), i \in [n]\}$ be a collection of $n$ bags of size $k$.[7] Let $\sup_{\widehat{y}, y} \ell(\widehat{y}, y) \leq B$, and $C_n^k = 2\left(\sqrt{2k \log(kn)} + 1\right)$. Then the following bound holds uniformly for all $h \in \mathcal{H}$ with probability at least $1 - \delta$:*

$$\left| \mathcal{L}(h) - \frac{1}{nk} \sum_{i,j} \widetilde{\ell}(h(x_{ij}), \alpha_i) \right| \leq C_n^k \left( \mathfrak{R}_{kn}(\mathcal{H}_\ell^{(1)}) + \mathfrak{R}_{kn}(\mathcal{H}_\ell^{(0)}) \right) + \frac{4B}{n} + 4B\sqrt{\frac{k \log(2/\delta)}{2n}},$$

*where $\mathcal{H}_\ell^{(1)} = \{x \to \ell(h(x), 1) \colon h \in \mathcal{H}\}$ and $\mathcal{H}_\ell^{(0)} = \{x \to \ell(h(x), 0) \colon h \in \mathcal{H}\}$.*

*Proof.* For $h \in \mathcal{H}$, bag $\mathcal{B}$ and label proportion $\alpha$ define

$$L(h, \mathcal{B}, \alpha) = \frac{1}{k} \sum_{x \in \mathcal{B}} \widetilde{\ell}(h(x), \alpha).$$

Let $\Phi(S) = \sup_{h \in \mathcal{H}} \mathbb{E}[\ell(h(x), y)] - \frac{1}{n} \sum_{i=1}^n L(h, \mathcal{B}_i, \alpha_i)$ .

Let $(x_{ij}, y_{ij})_{i \in [n], j \in [k]}$ denote the *instance* level sample. Let $\mathcal{S}' = (\mathcal{B}'_i, \alpha'_i)_{i \in [k]}$ denote the sample obtained by switching a single sample $(x_{ij}, y_{ij})$ to $(x'_{ij}, y'_{ij})$. Without loss of generality assume we switch sample $(x_{11}, y_{11})$. Then by the subadditive property of the supremum and the fact that $(\mathcal{B}_i, \alpha_i) = (\mathcal{B}'_i, \alpha'_i)$ for $i \neq 1$ we have

$$\Phi(\mathcal{S}) - \Phi(\mathcal{S}') \leq \frac{1}{n} \sup_{h \in \mathcal{H}} L(h, \mathcal{B}_1, \alpha_1) - L(h, \mathcal{B}'_1, \alpha'_1).$$

If we expand the difference inside the supremum for a fixed $h$ we have:

$$|L(h, \mathcal{B}_1, \alpha_1) - L(h, \mathcal{B}'_1, \alpha'_1)| \leq \frac{1}{k} \sum_{j=1}^k |\widetilde{\ell}(h(x_{1j}), \alpha_1) - \widetilde{\ell}(h(x'_{1j}), \alpha'_1)| .$$

Now using the fact that $|\alpha_1 - \alpha'_1| \leq \frac{1}{k}$ — as only one label changed — and $\widetilde{\ell}$ is $2kB$-Lipchitz as a function of $\alpha$ we must have, for $j \neq 1$,

$$|\widetilde{\ell}(h(x_{1j}), \alpha_1) - \widetilde{\ell}(h(x'_{1j}), \alpha'_1)| = |\widetilde{\ell}(h(x_{1j}), \alpha_1) - \widetilde{\ell}(h(x_{1j}), \alpha'_1)| \leq 2B .$$

On the other hand, using the fact that $k|\alpha - p| + \max(p, 1 - p) \leq k + 1$, and again the fact that $\widetilde{\ell}$ is Lipchitz with respect to $\alpha$ we see that:

$$|\widetilde{\ell}(h(x_{11}), \alpha_1) - \widetilde{\ell}(h(x'_{11}), \alpha'_1)| \leq (k + 1)B + 2B = B(k + 3).$$

---

[7]Our results can easily be extended to the case where bags have different sizes, provided the items are still i.i.d. The equal size assumption is made here mainly for notational convenience. If the bag sizes are constant with the number of bags $n$, then a rate of the form $1/\sqrt{n}$ can still be obtained. On the contrary, if the bag sizes scale with $n$, the actual rate will depend on the interplay between the bag sizes and $n$ itself.

We thus have that

$$|L(h, B_1, \alpha_1) - L(h, B_1', \alpha_1')| \leq \frac{1}{k}\left(B(k+3) + 2(k-1)B\right) = \frac{(3k+1)}{k}B \leq 4B$$

We therefore have

$$\Phi(\mathcal{S}) - \Phi(\mathcal{S}') \leq \frac{4B}{n} .$$

By McDiarmid's inequality and the fact that we have $kn$ individual samples we thus have that with probability at least $1 - \delta$:

$$\Phi(\mathcal{S}) \leq \mathbb{E}[\Phi(\mathcal{S})] + 4B\sqrt{\frac{k\log(1/\delta)}{2n}}$$

We proceed to bound the expectation of $\Phi(S)$:

$$\mathbb{E}[\Phi(\mathcal{S})] = \mathbb{E}_{\mathcal{S}}\left[\sup_{h \in \mathcal{H}} \mathbb{E}[\ell(h(x), y)] - \frac{1}{n}\sum_{i=1}^{n} L(h, \mathcal{B}_i, \alpha_i)\right]$$

$$= \mathbb{E}_{\mathcal{S}}\left[\sup_{h \in \mathcal{H}} \mathbb{E}_{\mathcal{S}'}\left[\frac{1}{n}\sum_{i=1}^{n} L(h, \mathcal{B}_i', \alpha_i')\right] - L(h, \mathcal{B}_i, \alpha_i)\right]$$

$$\leq \frac{1}{n}\mathbb{E}_{\mathcal{S},\mathcal{S}'}\left[\sup_{h \in \mathcal{H}} \sum_{i=1}^{n} L(h, \mathcal{B}_i', \alpha_i') - L(h, \mathcal{B}_i, \alpha_i)\right] ,$$

where $\mathcal{S}'$ is another i.i.d. sample drawn from the same distribution as $\mathcal{S}$. The second inequality follows from the fact that $\mathbb{E}[L(h, \mathcal{B}, \alpha)] = \mathbb{E}[\ell(h(x), y)]$. Let $\sigma_{ij}$ be a random variable uniformly distributed in $\{-1, 1\}$ and $\boldsymbol{\sigma} = (\sigma_{ij})_{i \in [n], j \in [k]}$. Then by a standard Rademacher complexity argument (e.g., [18]), we have

$$\frac{1}{n}\mathbb{E}_{\mathcal{S},\mathcal{S}'}\left[\sup_{h \in \mathcal{H}} \sum_{i=1}^{n} L(h, \mathcal{B}_i'\alpha_i') - L(h, \mathcal{B}_i, \alpha_i)\right] = \frac{1}{kn}\mathbb{E}_{\mathcal{S},\mathcal{S}'}\left[\sup_{h \in \mathcal{H}} \sum_{i,j} \widetilde{\ell}(h(x_{ij}'), \alpha_i') - \widetilde{\ell}(h(x_{ij}), \alpha_i)\right]$$

$$= \frac{2}{kn}\mathbb{E}_{\mathcal{S},\boldsymbol{\sigma}}\left[\sup_{h \in \mathcal{H}} \sum_{i,j} \sigma_{ij}\widetilde{\ell}(h(x_{ij}), \alpha_i)\right] . \quad (12)$$

Let $\beta > 0$ and $\eta = \frac{\beta}{kn}$. By Proposition B.5 we know that we can rewrite $\widetilde{\ell}(h(x_{ij}), \alpha_i)$ as $\widetilde{\ell}_\eta(h(x_{ij}), \alpha_i) + Z_{ijh}$. Moreover $|Z_{ijh}| \leq 2kB$ and by the union bound, with probability at least $1 - \beta$, for all $i, j, h$ we have $Z_{ijh} = 0$. Using this fact we may bound (12) as follows

$$\frac{2}{kn}\mathbb{E}_{\mathcal{S},\boldsymbol{\sigma}}\left[\sup_{h \in \mathcal{H}} \sum_{i,j} \sigma_{ij}\widetilde{\ell}(h(x_{ij}), \alpha_i)\right] \leq \frac{2}{kn}\mathbb{E}_{\mathcal{S},\boldsymbol{\sigma}}\left[\sum_{i,j} \sigma_{ij}\widetilde{\ell}_\eta(h(x_{ij}), \alpha_i)\right] + \frac{2}{kn}\mathbb{E}_{\mathcal{S},\boldsymbol{\sigma}}\left[\sup_{h \in \mathcal{H}} \sum_{i,j} \sigma_{ij}Z_{ijh}\right]$$

$$\leq \frac{2}{kn}\mathbb{E}_{\mathcal{S},\boldsymbol{\sigma}}\left[\sum_{i,j} \sigma_{ij}\widetilde{\ell}_\eta(h(x_{ij}), \alpha_i)\right] + 4kB\beta . \quad (13)$$

Finally, we use the definition of $\widetilde{\ell}_\eta$ to bound the above expression:

$$\widetilde{\ell}_\eta(h(x_{ij}), \alpha_i) = (k(\alpha_i - p) + p)\,\mathbb{1}_{|\alpha_i - p| < c(k,\eta)}\ell(h(x_{ij}), 1) + (k(p - \alpha_i) + (1-p))\,\mathbb{1}_{|\alpha_i - p| < c(k,\eta)}\ell(h(x_{ij}), 0)$$

$$= \psi_i^{(1)}\big(\ell(h(x_{ij}, 1)\big) + \psi_i^{(0)}\big(\ell(h(x_{ij}, 1)\big)$$

where $\psi_i^{(1)}(z) = (k(\alpha_i - p) + p)\,\mathbb{1}_{|\alpha_i - p| < c(k,\eta)}$ and $\psi_i^{(0)}$ is similarly defined. Notice that due to the indicator function we must have that $\psi_i^{(r)}$ is $(kc(k,\eta) + 1)$-Lipchitz. Define the set of functions

$\mathcal{H}_\ell^{(1)} = \{x \to \ell(h(x), 1) \colon h \in \mathcal{H}\}$ and $\mathcal{H}_\ell^{(0)} = \{x \to \ell(h(x), 0) \colon h \in \mathcal{H}\}$ we then have that

$$\frac{2}{kn} \mathop{\mathbb{E}}_{\mathcal{S},\boldsymbol{\sigma}} \left[ \sum_{i,j} \sigma_{ij} \widetilde{\ell}_\eta(h(x_{ij}), \alpha_i) \right]$$

$$\leq \frac{2}{kn} \mathop{\mathbb{E}}_{\mathcal{S},\boldsymbol{\sigma}} \left[ \sup_{h \in \mathcal{H}} \sum_{i,j} \sigma_{ij} \psi_i^{(1)} \big(\ell(h(x_{ij}), 1)\big) \right] + \frac{2}{kn} \mathop{\mathbb{E}}_{\mathcal{S},\boldsymbol{\sigma}} \left[ \sup_{h \in \mathcal{H}} \sum_{i,j} \sigma_{ij} \psi_i^{(0)} \big(\ell(h(x_{ij}), 0)\big) \right]$$

$$= \frac{2}{kn} \mathop{\mathbb{E}}_{\mathcal{S},\boldsymbol{\sigma}} \left[ \sup_{g \in \mathcal{H}_\ell^{(1)}} \sum_{i,j} \sigma_{ij} \psi_i^{(1)} \circ g(x_{ij}) \right] + \frac{2}{kn} \mathop{\mathbb{E}}_{\mathcal{S},\boldsymbol{\sigma}} \left[ \sup_{g \in \mathcal{H}_\ell^{(0)}} \sum_{i,j} \sigma_{ij} \psi_i^{(0)} \circ g(x_{ij}) \right] .$$

Finally, we use the fact that all functions $\psi_i^{(r)}$ are $(kc(k,\eta)+1)$-Lipchitz together with Talagrand's contraction Lemma [14, 18] to show that the above expression is bounded by

$$2\big(kc(k,\eta)+1\big) \left( \mathfrak{R}_{kn}(\mathcal{H}_\ell^{(1)}) + \mathfrak{R}_{kn}(\mathcal{H}_\ell^{(0)}) \right) = 2 \left( \sqrt{k \log \frac{1}{\eta}} + 1 \right) \left( \mathfrak{R}_{kn}(\mathcal{H}_\ell^{(1)}) + \mathfrak{R}_{kn}(\mathcal{H}_\ell^{(0)}) \right)$$

$$= 2 \left( \sqrt{k \log \frac{kn}{\beta}} + 1 \right) \left( \mathfrak{R}_{kn}(\mathcal{H}_\ell^{(1)}) + \mathfrak{R}_{kn}(\mathcal{H}_\ell^{(0)}) \right) .$$

Replacing this expression in (13) and setting $\beta = \frac{1}{kn}$ we see that

$$\mathbb{E}[\Phi(\mathcal{S})] \leq 2 \left( \sqrt{2k \log(kn)} + 1 \right) \left( \mathfrak{R}_{kn}(\mathcal{H}_\ell^{(1)}) + \mathfrak{R}_{kn}(\mathcal{H}_\ell^{(0)}) \right) + \frac{4B}{n} .$$

Putting it all together we see that with probability at least $1 - \delta$ over $\mathcal{S}$ we have:

$$\Phi(\mathcal{S}) \leq 2 \left( \sqrt{2k \log(kn)} + 1 \right) \left( \mathfrak{R}_{kn}(\mathcal{H}_\ell^{(1)}) + \mathfrak{R}_{kn}(\mathcal{H}_\ell^{(0)}) \right) + \frac{4B}{n} + 4B \sqrt{\frac{k \log(1/\delta)}{2n}}.$$

The result follows from the definition of $\Phi$. $\qquad\square$

### B.5 Supplementary material to Section 6

---
**Algorithm 1** SGD Using Pick-One from Each Bag
---
1: **Input** $\mathcal{S} = \{(\mathcal{B}_i, \alpha_i), i \in [n]\}, \{\eta_i : i \in [n]\}, \mathbf{w}_0$
2: $\mathbf{w} \leftarrow \mathbf{w}_0$
3: **for** $t \in \{1, \ldots, n\}$ **do**
4:     Pick $j$ from $[k]$ uniformly at random
5:     Update the model parameter $\mathbf{w}_t$ as

$$\mathbf{w}_{t+1} \leftarrow \Pi_{\mathcal{W}} \big( \mathbf{w}_t - \eta_i \widetilde{g}_{\mathbf{w}_t}(x_{tj}, \alpha_t) \big)$$

    with $\widetilde{g}_{\mathbf{w}_t}(x_{tj}, \alpha_t)$ defined in (3)
6: **end for**
7: **Return** $\mathbf{w}_{n+1}$

---

**Theorem 6.1.** *Suppose that $F(\mathbf{w}) = \mathbb{E}[\ell_\mathbf{w}(x, y)]$ is convex, $\mathbb{E}[\|g_{\mathbf{w}_t}(x, y)\|^2] \leq G^2$ for all $t \in [n]$, and $\sup_{\mathbf{w}, \mathbf{w}' \in \mathcal{W}} \|\mathbf{w} - \mathbf{w}'\| \leq D$. Consider Algorithm 1, run with step size $\eta_t = 1/\sqrt{k \cdot t}$. Then for any $n > 1$ we have*

$$\mathbb{E}[F(\mathbf{w}_n) - F(\mathbf{w}^*)] \leq \sqrt{k} \cdot \big( D^2 + 5G^2 \big) \frac{2 + \log(n)}{\sqrt{n}} .$$

*Proof.* We focus on the sequence of parameter vectors $\mathbf{w}_0, \ldots, \mathbf{w}_n$, which is a stochastic process, the filtration of which is defined as

$$\mathcal{F}_t = \{(x_{11}, y_{11}), \ldots, (x_{tj}, y_{tj}), J_1, \ldots, J_{t-1}\}$$

such that $J_i$ is a uniform random variable from $[k]$ for any $i$. We introduce some short-hand notations: the unbiased gradient based on surrogate labels is denoted by $\widetilde{g}_{ti} = \widetilde{g}_{w_t}(x_{tj}, \alpha_t)$ and $\widetilde{g}_t = \widetilde{g}_{tJ_t}$ is the gradient that is used by Algorithm 1, i.e., the unbiased gradient compute based on a randomly selected instance from $\mathcal{B}_t$. Furthermore we will denote the instance level gradient as $g_{tj} = g_{w_t}(x_{tj}, y_{tj})$. We can rewrite the L2 error as follows:

$$
\begin{aligned}
\mathbb{E}\left[\|\mathbf{w}_{t+1} - \mathbf{w}\|^2 |\mathcal{F}_t\right] &= \mathbb{E}\left[\left\|\Pi_{\mathcal{W}}\left(\mathbf{w}_t - \eta_t \widetilde{g}_t - \mathbf{w}\right)\right\|^2 |\mathcal{F}_t\right] \\
&\leq \mathbb{E}\left[\|\mathbf{w}_t - \eta_t \widetilde{g}_t - \mathbf{w}\|^2 |\mathcal{F}_t\right] \\
&\leq \mathbb{E}\left[\|\mathbf{w}_t - \mathbf{w}\|^2 |\mathcal{F}_t\right] - 2\eta_t \mathbb{E}\left[\langle \widetilde{g}_t, \mathbf{w}_t - \mathbf{w}\rangle |\mathcal{F}_t\right] + \eta_t^2 \mathbb{E}\left[\|\widetilde{g}_t\|^2 |\mathcal{F}_t\right] \\
&= \mathbb{E}\left[\|\mathbf{w}_t - \mathbf{w}\|^2 |\mathcal{F}_t\right] - 2\eta_t \frac{1}{k}\sum_{j=1}^k \mathbb{E}\left[\langle g_{tj}, \mathbf{w}_t - \mathbf{w}\rangle |\mathcal{F}_t\right] + \eta_t^2 \mathbb{E}\left[\|\widetilde{g}_t\|^2 |\mathcal{F}_t\right]
\end{aligned}
$$

$$(14)$$

where the last step follows from Proposition 4.2 applied to each term of the summation by noting that $\mathbf{w}_t - \mathbf{w}$ is not a random quantity with respect to filtration $\mathcal{F}_t$. As a next step, we will upper bound using the assumption that $\mathbb{E}[\|g_{\mathbf{w}_t}(x,y)\|^2] \leq G^2$ for any $t$ as

$$
\mathbb{E}\left[\|\widetilde{g}_t\|^2 |\mathcal{F}_t\right] = \mathbb{E}\left[\|\widetilde{g}_{tJ_t}\|^2 |\mathcal{F}_t\right] = \frac{1}{k}\sum_{j=1}^k \mathbb{E}\left[\|\widetilde{g}_{i,j}\|^2 |\mathcal{F}_t, J_t = j\right]
$$

so we shall focus on $\mathbb{E}\left[\|\widetilde{g}_{\mathbf{w}}(x, \alpha)\|^2\right]$ assuming that $\mathbf{w}$ is fixed. Based on Lemma B.2, we can compute an upper bound $\mathbb{E}\left[\|\widetilde{g}_{\mathbf{w}}(x, \alpha)\|^2\right]$ as

$$
\begin{aligned}
\mathbb{E}\left[\|\widetilde{g}_{\mathbf{w}}(x, \alpha)\|^2\right] &= \mathbb{E}\left[\|(k(\alpha - p) + p)g_{\mathbf{w}}(x, 1) + (k(p - \alpha) + (1 - p))g_{\mathbf{w}}(x, 0)\|^2\right] \\
&= \mathbb{E}\left[\|(k(\alpha - p) + p)g_{\mathbf{w}}(x, 1)\|^2\right] \\
&\quad + 2\mathbb{E}\left[\langle (k(\alpha - p) + p)g_{\mathbf{w}}(x, 1), (k(p - \alpha) + (1 - p))g_{\mathbf{w}}(x, 0)\rangle\right] \\
&\quad + \mathbb{E}\left[\|(k(p - \alpha) + (1 - p))g_{\mathbf{w}}(x, 0)\|^2\right] \\
&\leq 2k^2 G^2 + 2k^2 \mathbb{E}\left[\langle g_{\mathbf{w}}(x, 1), g_{\mathbf{w}}(x, 0)\rangle\right] \\
&\leq 4k^2 G^2
\end{aligned}
$$

where the last inequality follows from the Cauchy-Schwartz inequality applied to

$$
|\langle g_{\mathbf{w}}(x, 1), g_{\mathbf{w}}(x, 0)\rangle| \leq \|g_{\mathbf{w}}(x, 1)\| \cdot \|g_{\mathbf{w}}(x, 0)\| \leq G^2
$$

and the fact that $p(1 - p) \leq 1/4$. The convexity of the loss and (14) yield that

$$
\begin{aligned}
\mathbb{E}\left[F(\mathbf{w}_t) - F(\mathbf{w})|\mathcal{F}_t\right] &\leq \frac{1}{k}\sum_{j=1}^k \mathbb{E}\left[\langle g_{tj}, \mathbf{w}_t - \mathbf{w}\rangle |\mathcal{F}_t\right] \\
&\leq \frac{\mathbb{E}\left[\|\mathbf{w}_t - \mathbf{w}\|^2 |\mathcal{F}_t\right] - \mathbb{E}\left[\|\mathbf{w}_{t+1} - \mathbf{w}\|^2 |\mathcal{F}_t\right]}{2\eta_t} + \frac{\eta_t}{2}\mathbb{E}\left[\|\widetilde{g}_t\|^2 |\mathcal{F}_t\right] \\
&\leq \frac{\mathbb{E}\left[\|\mathbf{w}_t - \mathbf{w}\|^2 |\mathcal{F}_t\right] - \mathbb{E}\left[\|\mathbf{w}_{t+1} - \mathbf{w}\|^2 |\mathcal{F}_t\right]}{2\eta_t} + \frac{5\eta_t k^2 G^2}{2}
\end{aligned}
$$

which can be applied recursively as

$$
\mathbb{E}\left[\sum_{t=n-s}^n (F(\mathbf{w}_t) - F(\mathbf{w}))\Big|\mathcal{F}_t\right] \leq \frac{1}{2\eta_{n-s}}\mathbb{E}\left[\|\mathbf{w}_{n-s} - \mathbf{w}\|^2 |\mathcal{F}_{n-s}\right]
$$

$$+ \sum_{t=n-s+1}^{n} \frac{\mathbb{E}\left[\|\mathbf{w}_t - \mathbf{w}\|^2 \,|\, \mathcal{F}_t\right]}{2} \left(\frac{1}{\eta_t} - \frac{1}{\eta_{t-1}}\right) + \frac{5G^2 k^2}{2} \sum_{t=n-s}^{n} \eta_t \,.$$

Now we can follow Theorem 2 of [27] with $\eta = c/(k \cdot \sqrt{t})$ where the index $t$ is meant over bags. Let us upper bound $\mathbb{E}[\|\mathbf{w}_t - \mathbf{w}\|^2]$ by $D^2$ and pick $\mathbf{w} = \mathbf{w}_{n-s}$ which yields

$$\mathbb{E}\left[\sum_{t=n-s}^{n} (F(\mathbf{w}_t) - F(\mathbf{w}_{n-s}))\Big| \mathcal{F}_t\right] \leq \frac{kD^2}{2c}\left(\sqrt{n} - \sqrt{n-s-1}\right) + \frac{5ckG^2}{2} \sum_{t=n-s}^{n} \frac{1}{\sqrt{t}}$$

$$\leq \left(\frac{kD^2}{2c} + 5ckG^2\right)\left(\sqrt{n} - \sqrt{n-s-1}\right)$$

$$\leq \left(\frac{kD^2}{2c} + 5ckG^2\right) \frac{s+1}{\left(\sqrt{n} + \sqrt{n-s-1}\right)}$$

$$\leq \left(\frac{kD^2}{2c} + 5ckG^2\right) \frac{s+1}{\sqrt{T}}$$

where we used the fact that $\sum_{t=n-s}^{n} 1/\sqrt{t} \leq 2(\sqrt{n} - \sqrt{n-s-1})$. The rest of the proof is analogous to the proof of Theorem 2 of [27]. $\qquad \square$

## C  Extension to Multiclass Classification

In the main body of the paper have restricted our analysis to binary classification problems. This was mostly done for ease of exposition. We now show that EASYLLPcan easily generalize to the multi-class scenario. This is in contrast to other methods like that of [8] whose generalization to multi-class requires solving an optimization problem for every gradient step. For the rest of this section we let $C$ denote the number of classes and $\mathcal{Y} = \{1, \ldots, C\}$ denote the label space. The result below is a generalization of its binary counterpart.

**Theorem C.1.** *Let $k > 0$ and $(x_1, y_1), \ldots, (x, y_k)$ be a sample from distribution $\mathcal{D}$. For each class $c$, let $\alpha_c = \frac{1}{k}\sum_{i=1}^{k} \mathbb{1}_{y_i=c}$ denote the fraction samples with label $c$. Let $\mathcal{B} = \{x_1, \ldots, x_k\}$ and define $\boldsymbol{\alpha} = (\alpha_1, \ldots, \alpha_k)$. Finally, let $p_c = \mathbb{P}(y = c)$. Let $g\colon \mathcal{X} \times \mathcal{Y} \to \mathbb{R}^d$ be any (measurable) function over features and labels. Define the soft label corrected function $\widetilde{g}$ as:*

$$\widetilde{g}(x, \boldsymbol{\alpha}) = \sum_{c=1}^{C} \big(k\alpha_c - (k-1)p_c\big)g(x, c).$$

*For any $x_i \in \mathcal{B}$, the soft label corrected function satisfies: $\mathbb{E}[\widetilde{g}(x_i, \boldsymbol{\alpha})] = \mathbb{E}_{x,y\sim\mathcal{D}}[g(x, y)]\,.$*

*Proof.* Fix $i \in [k]$ and define $\alpha_c^{(i)} = \alpha_c - \frac{1}{k}\mathbb{1}_{y_i=c}$ so that $\mathbb{1}_{y_i=c} = k(\alpha_c - \alpha_c^{(i)})$. Moreover, since $\alpha_c^{(i)} = \frac{1}{k}\sum_{j\neq i} \mathbb{1}_{y_j=c}$ and the samples $(x_1, y_1), \ldots, (x_k, y_k)$ are independent, we are guaranteed that $\alpha_c^{(i)}$ is independent of $x_i$. Let $x_i \in \mathcal{B}$ and let $y_i$ be its corresponding label we then have

$$\mathbb{E}[g(x_i, y_i)] = \sum_{c=1}^{C} \mathbb{E}[\mathbb{1}_{y_i=c}\, g(x_i, c)] = \sum_{c=1}^{C} \mathbb{E}[k(\alpha_c - \alpha_c^{(i)})g(x_i, c)]$$

$$= \sum_{c=1}^{C} \mathbb{E}[k\alpha_c\, g(x_i, c)] - \sum_{c=1}^{C} \mathbb{E}[k\alpha_c^{(i)}\, g(x_i, c)]\,.$$

Now, since $\alpha_c^{(i)}$ is independent of $g(x_i, c)$ and $\mathbb{E}[\alpha_c^{(i)}] = \frac{k-1}{k}p_c$, we have that $\mathbb{E}[k\alpha_c^{(i)}g(x_i, c)] = (k-1)p_c\, \mathbb{E}[g(x_i, c)]$. Therefore the above expression can be simplified as

$$\mathbb{E}_{x,y\sim\mathcal{D}}[g(x, y)] = \mathbb{E}[g(x_i, y_i)] = \sum_{c=1}^{C} \mathbb{E}[k\alpha_c g(x_i, c)] - \sum_{c=1}^{C}(k-1)p_c\, \mathbb{E}[g(x_i, c)]$$

$$= \mathbb{E}\left[\sum_{c=1}^{C}\big(k\alpha_c - (k-1)p_c\big)g(x_i, c)\right] = \mathbb{E}[\widetilde{g}(x_i, \boldsymbol{\alpha})]\,,$$

as claimed. $\qquad\square$