# OpenReview forum: "Easy Learning from Label Proportions"
_NeurIPS.cc/2023/Conference — NeurIPS 2023 poster_

### Official Review · Reviewer_YhoX · 2023-06-29

**Soundness:** 3 good
**Presentation:** 3 good
**Contribution:** 3 good
**Rating:** 6
**Confidence:** 4

**Summary:**

This paper focuses on the problem of learning from label proportions. It addresses the performance degradation issue of the EPRM method when the hypothesis class lacks sufficient expressiveness. To overcome this problem, the paper introduces EasyLLP as a solution. In practice, EasyLLP differs from EPRM (or PropMatching) in the calculation process. EasyLLP first calculates the corrected instance-level loss and then takes the average, whereas EPRM first takes the average of the bag-level predictions and then calculates the loss. This simplifies the implementation of the learning algorithm: just utilize a corrected loss function and approach the problem as a regression task.






**Strengths:**

This paper studies the problem of LLP, which is an important problem to the community. The proposed method is based on a novel loss correction method, which could be useful for some related problems.

The paper points out and analyzes the limitation of EPRM. This finding is interesting and novel.

The proposition 4.2 with its proof seem to be a significant contribution. Based on proposition 4.2, the proposed EasyLLP is sound and easy to implement. Experimental results show the effectiveness of the proposed EasyLLP.

**Weaknesses:**

There are some minor issues:
1. Literature [16] is also a method based on unbiased risk estimation, and is very related to the proposed method. How's the comparison between EasyLLP and [16]? It should be compared and discussed in the experiments.
2. In section 3 the paper discussed the limitation of EPRM. However, why EasyLLP solves this problem is unclear. Please add some discussion.

**Questions:**

See weaknesses part.

---

> ### Author Rebuttal · Authors · 2023-08-09
>
> > On originality: Comparison to [16], and other papers in this literature.
>
> * A similar point was raised by Reviewer c8Fj. Thanks for stimulating a more thorough discussion of the related literature. The paper Reviewer YhoX is mentioning is related to ours, but it is by no means subsuming our results. For instance, [16] (as well as other papers in this stream of literature), rely on two or more $U$ sets, which are assumed to be diverse in the prior mixture. It is this diversity that allows the authors to construct unbiased estimates and then derive consistency results. In our case, the bags have the same prior $p$, and we work under the assumption that we cannot handcraft diverse bags out of our samples, as the aggregation into bags is done without having access to the class conditional distributions $p(x|y=1)$ and $p(x|y=-1)$. This setting is largely motivated by practical scenarios where the learner may not have control over the way bags are generated. The difference between the two settings can also be observed in the different flavor of the consistency results. E.g., in [16] (even with $m = 2$ bags) the consistency limit has to be interpreted ``as the bag size $n_{tr} = n_1+n_2\ $ goes to infinity”. In our case, the bag size $k$ has to remain constant, and it is the number of bags that goes to infinity.
>
> * One more thing that we also found different is that, unlike our paper, all results in these previous works, as presented, make assumptions about the loss function (e.g., proper losses like square loss or cross entropy for [16], margin-based losses for Lu et al. “On the minimal supervision” ICLR 2019 paper). This flexibility allows us to apply our debiasing procedure to any function g(x,y) of two variables, hence we can debias, e.g., also the *gradient* of a loss function, enabling the principled usage of stochastic gradient descent procedures with only label proportion information. This is something that comes for free in our approach, that we have not seen in the literature. Moreover, in Appendix C, we have an extension to the multiclass case.
>
> * We will add more detailed discussion in the related work section of the paper.
>
>
> > Limitations of EPRM and why EasyLLP overcomes them.
>
> The main difference between EPRM (aka Proportion Matching) and EasyLLP seems to be that the former can be catastrophically bad in non realizable settings (i.e., when the function $h^* : x \mapsto P(y=1\mid x)$ is not an element of the function class $\mathcal{H}$). On the other hand, EasyLLP is not relying on this assumption. When confronted with a non realizable setting, the EasyLLP solution will simply converge to the best in class solution within class $\mathcal{H}$. Yet, it should be added that Thm 3.2 and Cor 3.3 only provide *sufficient* conditions for EPRM to perform well. We do not know to what extent these conditions are also necessary, and in lines 192-201 we give an example (only verified empirically) where, if these conditions are not satisfied, the EPRM minimization criterion can cause learning to go completely off trail.

---

> > ### Comment · Reviewer_YhoX · 2023-08-15
> >
> > Thanks for your response. I have read the response and other reviews.
> > I was expecting an experimental comparison with [16].
> > I will keep the score as is.

---

> > > ### Author Response · Authors · 2023-08-18
> > > **On the experimental comparison to [16] (and related papers)**
> > >
> > > Thank you for the response.
> > >
> > > Due to the differences in problem setup between our work and that of [16], it is unclear how a fair comparison should be performed. In particular, in the third paragraph of section 4, the authors of [16] write: “Note that in most LLP papers, each $U$ set is uniformly sampled from the shuffled $U$ training data, therefore the label proportions of all $U$ sets are the same in expectation. As the set size increases, all the proportions converge to the same class prior, making the LLP problem computationally intractable. As shown above, our experimental scheme avoids this issue by determining valid class priors before sampling each $U$ set.”
> > >
> > > Our work is in the LLP setting the authors describe where bags contain i.i.d. examples and all bags have the same class prior. As we have been trying to emphasize in our first response, this is quite different from the setup considered in the experiments of [16]. Instead, in [16] each bag ($U$ set) is assigned a random class prior $\pi$ from the interval $[0.1, 0.9]$, and then the $U$ set is filled with examples drawn from the mixture distribution given by $p_{tr}(x) = \pi p_p(x) + (1-\pi) p_n(x)$, where $p_p$ and $p_n$ are the conditional densities of $x$ for the positive and negative class, respectively. Our methods were not designed for the setting of [16], and the methods of [16] were not designed for our setting.

---

### Official Review · Reviewer_z2CW · 2023-07-07

**Soundness:** 3 good
**Presentation:** 3 good
**Contribution:** 3 good
**Rating:** 6
**Confidence:** 4

**Summary:**

The paper aims to advance the theoretical understanding behind the LLP problem, and provide the conditions under which the algorithm is expected to work. They propose a theoretically founded algorithm for learning from label proportions called EasyLLP. In particular, they have shown how to estimate the expected value of any function of (x, y) pairs from labeled data. They have also shown complexity guarantees for ERM and convergence guarantees for SGD.

The authors have also evaluated their proposed approach against PropMatch and 2 baseline models, on 4 datasets: MNIST, CIFAR-10, UCI Adult, and Higgs. For all the datasets, the authors have considered the corresponding binary classification variant, if they were multi-class.
The results show

**Strengths:**

The authors have clearly described their approach, and demonstrated the results against baselines on 4 datasets. Understanding the theoretical nature of LLP models is important given the application of the domain. The paper is a good step in that direction.

**Weaknesses:**

In order to improve the reproducibility of the approach, it will be helpful if the authors can either share their code or provide pseudocode for using EasyLLP.

The evaluation is limited to fairly broad datasets such as MNIST, CIFAR and Higgs. The LLP problem is applied to various real world problems, including those rightfully noted by the authors, such as advertising. The datasets in such domains have additional challenges such as a more complex feature space and class imbalance. It will be great if the results can be shown with such complexities taken into account.

I'd encourage the authors to consider more recent approaches in LLP for comparing their results.

**Questions:**

Concerns already listed under Weakness.

---

> ### Author Rebuttal · Authors · 2023-08-09
>
> > On reproducibility
>
> We find EasyLLP quite simple to reproduce, even without pseudocode. In any event, we will happily make the code for our experiments available.
>
> > Limited experimental evaluation
>
> We have strived to do as thorough an evaluation as possible by considering different datasets with different class proportions. Using advertising datasets can prove challenging, as there are no standard datasets we are aware of (for privacy and proprietary reasons) that can allow us to test the efficacy of our methods at predicting event level labels.
>
> > Comparison to more recent approaches in LLP
>
> We appreciate the suggestion and would find it very helpful if the reviewer could point us to the more recent approaches they are alluding to in their review.

---

> > ### Comment · Reviewer_z2CW · 2023-08-22
> >
> > @Authors: Thanks for addressing my concerns. Based on your response above and other reviwer responses, I am revising my score.

---

### Official Review · Reviewer_c8Fj · 2023-07-09

**Soundness:** 3 good
**Presentation:** 3 good
**Contribution:** 2 fair
**Rating:** 6
**Confidence:** 2

**Summary:**

The paper presents a debiasing approach called EASYLLP for Learning from Label Proportions (LLP), where only class label frequencies in bags are available. The authors provide theoretical analyses of a label proportion matching algorithm and propose a general debiasing technique for estimating instance loss. Experimental results demonstrate the effectiveness of their approach compared to existing methods in various learning frameworks.

**Strengths:**

The proposed method in the paper has several advantages. Firstly, it is described as simple and straightforward, making it easy to implement. This aspect can be beneficial for practitioners and researchers looking for an accessible solution to the problem of Learning from Label Proportions (LLP). Additionally, the paper is praised for being well-written, indicating clear and concise explanations of the concepts and methods presented.

**Weaknesses:**

My main concern is its relationship with previous works, such as [16] and references therein. Both of the papers assume data are generated at random, and propose a debiasing procedure via linear transformations, with similar theoretical results on consistency. It is difficult to see any fundamental innovation in the current work compared to previous works, especially given that all theoretical results are straightforward after confirming the consistency of the proposed risk.


For [16] and a more fundamental work On the Minimal Supervision for Training Any Binary Classifier from Only Unlabeled Data (ICLR19), they require separation of the class prior distribution, as argued in the paper, because they have a /theta-/theta' in the reweighting denominator, which could be reduced by multiplying them. The difference seems to be that the ICLR19 paper is calculating the expectation of risk using two sets, but the current paper is using one set; other derivatives are similar.

**Questions:**

Compare with [16] and other works such as On the Minimal Supervision for Training Any Binary Classifier from Only Unlabeled Data (ICLR19), what are the fundamental differences of the current paper?

---------------------
After rebuttal, I am satisfied with the answer on the two key differences, and would like to raise my score.

**Limitations:**

yes

---

> ### Author Rebuttal · Authors · 2023-08-09
>
> > On originality: Comparison to [16], and other papers like “On the Minimal Supervision…”
>
> * We thank the reviewer for stimulating a more thorough discussion of the related literature. The papers the reviewer is mentioning are related to ours, but they are by no means subsuming our results. For instance, both [16] and the “minimal supervision…” paper (as well as other papers in this stream of literature), rely on two or more $U$ sets, which are assumed to be diverse in the prior mixture. It is this diversity that allows the authors to construct unbiased estimates and then derive consistency results. In our case, the bags have the same prior $p$, and we work under the assumption that we cannot handcraft diverse bags out of our samples, as the aggregation into bags is done without having access to the class conditional distributions $p(x|y=1)$ and $p(x|y=-1)$. This setting is largely motivated by practical scenarios where the learner may not have any control on how bags are generated. The difference between the two settings can also be observed in the different flavor of the consistency results. E.g., in [16] (even with $m = 2$ bags) the consistency limit has to be interpreted ``as the bag size $n_{tr} = n_1+n_2\ $ goes to infinity”. In our case, the bag size $k$ has to remain constant, and it is the number of bags that goes to infinity.
>
> * One more thing that we also found different is that, unlike our paper, all results in these previous works, as presented, make assumptions about the loss function (e.g., square loss or cross entropy for [16], margin-based losses for the “minimal supervision…” paper). The fact that we make no assumptions allows us to apply our debiasing procedure to any function g(x,y) of two variables, hence we can debias, e.g., also the *gradient* of a loss function, enabling the principled usage of stochastic gradient descent procedures with only label proportion information. This is something that comes for free in our approach, that we have not seen in the literature. Moreover in Appendix C we have an extension to the multiclass case.
>
> * We will add more detailed discussion in the related work section of the paper.

---

### Official Review · Reviewer_xq8A · 2023-07-10

**Soundness:** 3 good
**Presentation:** 2 fair
**Contribution:** 3 good
**Rating:** 6
**Confidence:** 3

**Summary:**

The authors start by providing a theoretical analysis of the proportion matching algorithm, a standard algorithm from the literature that simply minimizes the loss over the average instance-level predictions.

**Strengths:**

I find the way the authors approach to the problem of learning from label proportions, and the goal they set out to achieve to be very worthwhile. I also believe the theoretical results therein might of interest to the community. I did, however, struggle at times in managing to follow the paper. Therefore, in my opinion, the authors really need to spend some effort polishing and refining the exposition to make it accessible to the broader community.

**Weaknesses:**

- My main gripe with the paper is that I would've liked to see more discussion, or intuition following each Theorem, corollary or proposition. For instance, I really would've liked the authors to spend some time discussing the assumption upon which Theorem 3.2 hinges, and if such an assumption is expected to hold in practice (although I do appreciate them giving an example for when it fails due to the model class not containing the true conditional distribution)

- Empirical evaluation seems to suggest improvement only with large bag sizes (2^7), and on some dataset not at all, compared to the baselines.


**Questions:**

- Could you please say more regarding the sentence starting line 166? While the objective being approximation by equation (1) is clear to me, I'm not clear own what it means for equation (1) to "correctly approximate" it.

- In Figure 1, left plot, why does the prop matching loss start out much lower compared to other methods?

- Could you please elaborate on the statement of Theorem 3.2? In particular, I've been trying to conclude how strong of an assumption it is that the minimizer of the expected loss the average of $Z$ (I do believe it is very strong, and seldom holds in any realistic setting)

- In definition 4.1:
  - Is $p$ as defined in the notation section or Theorem 3.2?
  - I find the use of $\alpha$ here confusing, as my first instinct was to think of it as a label proportion, but in the notation section you only define $\alpha$ as a function of a bag. I then noticed that it is any real-valued parameter in $\[0,1\]$.
  - Why does this definition make sense? I interpret this as somehow correcting the bias introduced by using bags instead of instances. If that is true, why does this correction term make sense?
  - I think this definition requires some notion of distribution due to the presence of $p$?

- In proposition 4.2:
  - the notation does not make it clear how $x_j$ is related to $\mathcal{B}$. I think something along the lines of $\mathbb{E}_{(x_j, \alpha) \sim (\mathcal{B}, \alpha)}$ would perhaps better convey your intent? But then I'm confused again, because somehow $\tilde{g}$ is a function of an individual instance and $\alpha$? But $\alpha$ is a function of the entire bag?
  - Am I correct in reading the equations as saying that the soft-label corrected function averaged over all bags and proportions is simply equal to $g$ averaged over the instance-level data distribution? If so, then it is my opinion that this needs be framed and discussed since, from what I understand, your entire approach hinges upon this result, and it is by no means easy to see.

- I'm very confused by Equation 5,

**Limitations:**

Adequately addressed

---

> ### Author Rebuttal · Authors · 2023-08-09
>
> We thank the reviewer for pointing out a number of places where further discussion and intuition would improve the presentation, and we will add additional discussion to the paper.
>
> > “Empirical evaluation seems to suggest improvement only with large bag sizes…”
>
> Yes, we agree that EasyLLP only sees empirical improvements for large bag sizes (and sometimes we do not outperform PropMatch at all). However, given that the LLP problem becomes more difficult as the bag size grows, we expect differences between methods to become more pronounced at larger bag sizes. In the complete set of experiments in appendix section A.5, EasyLLP is the only method that is consistently competitive with other methods at every bag size (e.g., in Figure 4 we see that PropMatch and DA have relatively poor performance even at bag size $k = 8$ for the two ConvNet models). We also know that there are cases (e.g., the synthetic example from Section 3) where PropMatch converges to a high loss model and will not improve even with access to more data.
>
> > “Could you please say more regarding the sentence starting line 166…”
>
> We mean that if you find a classifier that minimizes the empirical proportion matching loss (Equation 1), then as long as you have enough data, it will also approximately minimize the population level loss (Equation 2). We will clarify the language.
>
> > “In Figure 1, left plot…”
>
> Unlike EasyLLP, the proportion matching loss has a different meaning (i.e., it is a measure of how close the model’s average prediction on a bag is to the bag’s label proportion). On the other hand, EasyLLP is an estimate of the *event* training loss (not averaged over the bag), so we should expect the EasyLLP training loss to track the event training loss, but for the proportion matching loss to be somewhat unrelated. This behavior is indeed what we observe in Figure 1.
>
> > “Could you please elaborate on the statement of Theorem 3.2?”
>
> * Theorem 3.2 has two key assumptions: 1. The function $h^* : x \mapsto P(y=1\mid x)$ is an element of the hypothesis class $\mathcal{H}$ (realizability assumption) and 2. The loss function $\ell$ has the property that for any random variable $Z$, $E[Z]$ minimizes the function $\rho \mapsto E_Z[\ell(\rho, Z)]$.
>
> * The assumption about the loss is relatively mild, and in Corollary 3.2 (Line 181) we show that this holds for two commonly used losses: the binary cross-entropy and the squared losses.
>
> > **Questions about Definition 4.1:**
>
> > “Is $p$ as defined in the notation section or Theorem 3.2?”
>
> $p$ is as defined in the notation section, which is the marginal probability that $y = 1$. We will change the symbol used in Theorem 3.2 to avoid the conflict.
>
> > “I find the use of $\alpha$ here confusing…”
>
> The soft-label corrected function takes as input a feature vector $x$ and a label proportion $\alpha$. The intended interpretation is that $x$ will be a feature vector from a bag $B$ of examples, and alpha will be the label proportion (which is a random variable) for that bag. We have somehow overloaded the notation by viewing $\alpha$ as both a random variable (notation section) and its value (Def. 4.1).
>
> > “I think this definition requires some notion of distribution…”
>
> You are right, this definition depends on the value $p$, which is the marginal probability that $y = 1$ for the underlying data distribution. We will clarify this.
>
> > “Why does this definition make sense?”
>
> The soft-label corrected function $\tilde g$ is defined this way so that the subsequent Proposition 4.2 (unbiasedness) holds.
>
>
> > **Questions about Proposition 4.2**
>
> > “The notation does not make clear how $x_j$ is related to $B$”
>
> In the proposition there is a sample $(x_1, y_1), \dots, (x_k, y_k)$ drawn i.i.d. from the data distribution. Bag $B = (x_1, \dots, x_k)$ contains the feature vectors, and $\alpha = \frac{1}{k} \sum_{i=1}^k y_i$ is the proportion of the labels in the bag that are positive. The claim is that if we fix index $j \in [1, \ldots, k]\ $ and consider the $j$-th element $x_j$ in the bag, the expected value of $\tilde g(x_j, \alpha)$ (the expectation being w.r.t. the random draw of the labeled bag $(B,\alpha)$ or, equivalently, w.r.t.  the random sample $(x_1, y_1), \dots, (x_k, y_k)$) is equal to the expected value of $g(x,y)$, where the expectation is w.r.t. a fresh sample $(x,y)$ from the data distribution.
>
> > “But then I’m confused again because somehow $\tilde g$ is a function of an individual instance and $\alpha$?”
>
> Yes, $\tilde g$ is a function of one feature vector $x_j$ from the bag, and the label proportion $\alpha$ of the bag. So, in a sense, $\tilde g$ is a function of the entire sample $(x_1,y_1), \ldots, (x_k,y_k)$ (via $x_j$ and $\alpha$).
>
> > Am I correct in reading the equations as saying…”
>
> * Yes, your interpretation is essentially correct. For any data distribution $D$ over $(x,y)$ pairs, there is a corresponding distribution over bags with label proportions (i.e., $B = (x_1, …, x_k)$ and $\alpha$). Proposition 4.2 shows that it is possible to estimate the expected value of $g$ on the distribution $D$ given sample access to the bag and proportion distribution by using the soft-label corrected function. We will add more discussion to the paper to better elucidate the correct interpretation.
>
> * The paragraph on lines 252-260 was meant to outline how the soft-label corrected loss is applied in learning contexts, but we will certainly include some additional discussion following Proposition 4.2, as the reviewer suggests.
>
> > “I’m very confused by Equation 5”
>
> * Normally in empirical risk minimization, we find the classifier $h$ from a hypothesis space $\mathcal{H}$ that minimizes the loss on the training data. Equation 5 is the estimated training loss on a dataset of bags, where $x_{i j}$ is the $j$-th point of the $i$-th bag, and $\alpha_i$ is the label proportion for that bag.
>
> * The rest of Sect. 5 studies how much worse it is to minimize Equation 5 compared to the actual training loss.

---

> > ### Comment · Reviewer_xq8A · 2023-08-18
> >
> > Thank you for addressing my concerns. I have revised my score.

---

### Decision · Program_Chairs · 2023-09-21

**Decision:**

Accept (poster)

**Comment:**

All reviewers are positive on this work. I agree with the authors’ response regarding why they do not compare to [16]. Also convinced was at least one of the two reviewers raising this point. This paper is a joy to read, well-motivated, and progresses very naturally to the easy-to-implement (and very clearly defined, in my view) method EasyLLP. The authors have given good theoretical guarantees which avoid the need to assume that the infamous “Bayes-in-class” assumption. The technical results do not look to be overly sophisticated, but overall this is a nice work. Another advantage, noted by the authors in the discussion period, is that their work has the flexibility of transcending proper losses, unlike many previous works. I recommend this work to be accepted to the conference.